# Natural variants suppress mutations in hundreds of essential genes

Leopold Parts[1,2,3,*] (ID), Amandine Batté[4], Maykel Lopes[4], Michael W Yuen[1], Meredith Laver[1], Bryan-Joseph San Luis[1], Jia-Xing Yue[5] (ID), Carles Pons[6], Elise Eray[4], Patrick Aloy[6,7], Gianni Liti[5] (ID) & Jolanda van Leeuwen[4,**] (ID)

## Abstract

The consequence of a mutation can be influenced by the context in which it operates. For example, loss of gene function may be tolerated in one genetic background, and lethal in another. The extent to which mutant phenotypes are malleable, the architecture of modifiers and the identities of causal genes remain largely unknown. Here, we measure the fitness effects of ~ 1,100 temperature-sensitive alleles of yeast essential genes in the context of variation from ten different natural genetic backgrounds and map the modifiers for 19 combinations. Altogether, fitness defects for 149 of the 580 tested genes (26%) could be suppressed by genetic variation in at least one yeast strain. Suppression was generally driven by gain-of-function of a single, strong modifier gene, and involved both genes encoding complex or pathway partners suppressing specific temperature-sensitive alleles, as well as general modifiers altering the effect of many alleles. The emerging frequency of suppression and range of possible mechanisms suggest that a substantial fraction of monogenic diseases could be managed by modulating other gene products.

**Keywords** compensatory evolution; genetic interactions; genetic modifiers; genetic suppression; natural variation

**Subject Category** Genetics, Gene Therapy & Genetic Disease

**Mol Syst Biol. (2021) 17: e10138**

## Introduction

The phenotypic outcome of a mutation is determined by the genetic context in which it occurs. The causes of such variation are fascinating in themselves, but are also central to finding ways of predicting and ameliorating genetic diseases. Loss of gene function may lead to death of specific tumour cells only, making the gene a potent drug target (Behan *et al*, 2019; Gonçalves *et al*, 2020). Moreover, a coding mutation with no discernible impact in a parent can result in a disorder in their child (Wright *et al*, 2019). Understanding how such incomplete penetrance arises, and predicting it for a new context, would therefore deepen our understanding of cellular systems and likely impact diagnoses for developmental disorders or personalised treatments for tumours.

Viability is perhaps the simplest mutation phenotype to analyse. In the course of establishing the yeast gene knockout collection, it became clear that about 1,100 of the ~ 6,000 yeast genes are indispensable under standard, nutrient-rich growth conditions (Giaever *et al*, 2002). However, repeating this resource construction in another genetic background offered a tantalising glimpse into the complexity of mutant phenotypes, as over 5% of the essential genes were variable between two closely related strains (Dowell *et al*, 2010). New strain panels (Galardini *et al*, 2019; Sanchez *et al*, 2019) both established estimates of ~ 10% of all genes demonstrating variable knockout phenotypes between strains and species.

The reason for incomplete penetrance in general, and variable gene essentiality in particular, is the abundance of modifier loci that can suppress mutation effects (Hou *et al*, 2018). Although their existence has been appreciated for a century (Altenburg & Muller, 1920), validated examples remain elusive. A small number of modifiers have been mapped and validated for mouse models (Hamilton & Yu, 2012) and human disease (Harper *et al*, 2015; Riordan & Nadeau, 2017). By far, the most well studied are examples from yeast, powered by the availability of a large number of genetically diverged natural isolates (Peter *et al*, 2018), genetic tools that allow making large collections of loss-of-function alleles (Sanchez *et al*, 2019) and the ability to systematically cross strains in controlled designs (Tong *et al*, 2001; Hallin *et al*, 2016; Bloom *et al*, 2019).

1 Donnelly Centre for Cellular and Biomolecular Research, University of Toronto, Toronto, ON, Canada
2 Wellcome Sanger Institute, Wellcome Genome Campus, Hinxton, UK
3 Department of Computer Science, University of Tartu, Tartu, Estonia
4 Center for Integrative Genomics, University of Lausanne, Lausanne, Switzerland
5 University of Côte d'Azur, CNRS, INSERM, IRCAN, Nice, France
6 Institute for Research in Biomedicine (IRB Barcelona), The Barcelona Institute for Science and Technology, Barcelona, Spain
7 Institució Catalana de Recerca i Estudis Avançats (ICREA), Barcelona, Spain
  *Corresponding author. Tel: +44 1223 834 244; E-mail: leopold.parts@sanger.ac.uk
  **Corresponding author. Tel: +41 21 692 3920; E-mail: jolanda.vanleeuwen@unil.ch

Systematic identification of spontaneous mutations that can suppress fitness defects of "query" mutant alleles in a reference yeast strain has illuminated mechanisms of suppression (van Leeuwen *et al*, 2016, 2017, 2020). These studies have shown that although deletion mutants are mainly suppressed by genes with a role in the same functional module, partial loss-of-function alleles are frequently suppressed by more general mechanisms affecting query protein expression or stability. However, surveys in model organisms have been largely limited to detecting single gene suppression in a laboratory setting, whereas more complex networks of modifiers may affect the penetrance of any given allele in natural populations. Linkage-based analyses of large panels of individuals have indeed identified second and higher-order modifier effects (Chandler *et al*, 2014; Taylor & Ehrenreich, 2015; Mullis *et al*, 2018; Hou *et al*, 2019; Sanchez *et al*, 2019), but few modifiers are usually characterised in depth beyond mapping the loci in such designs. The relevance of established broad suppression mechanisms for natural populations thus remains unclear (Matsui *et al*, 2017).

Here, we measure phenotypes elicited by crossing about 1,100 temperature-sensitive mutant alleles of essential genes to ten genetically diverse yeast strains. We use powerful genetic mapping approaches to identify modifier loci of a subset and validate causal genes for 19 of them. A single strong suppressor allele could independently overcome the query mutation phenotype in nearly all mapped cases. The suppressing variants tend to operate within the same biological module as the query gene, with mutations in protein interaction partners or protein complexes often suppressing specific genes, mutations in pathways suppressing other pathway members and general modifiers altering the effect of many mutations. Together, these results demonstrate the natural genetic flexibility of

cells to fulfil crucial tasks and suggest that loss of human gene function could often be specifically rescued as well.

## Results

### Measuring suppression by standing variation

We set out to test mutation effects in segregant progeny from diverse yeast isolates from various geographic locations and sources ("wild yeasts") (Liti *et al*, 2009; Bergström *et al*, 2014). To do so, we used the Synthetic Genetic Array approach (Tong *et al*, 2001) to cross a collection of 1,106 temperature-sensitive alleles ("TS alleles") of 580 essential query genes in the laboratory strain S288C (Costanzo *et al*, 2016) to 10 stable haploid wild yeasts (Fig 1A) (Cubillos *et al*, 2009), as well as into the S288C control as a reference. We isolated pools of ~ 60,000 segregant progeny carrying the TS allele to obtain populations of haploid individuals with genomes that, except for the genomic regions around the TS allele and selection markers, are a mosaic of the reference and wild parents (Fig 1B, Methods). We grew the segregant pools at permissive (26°C) and restrictive (34°C) temperatures and measured their fitness. In the control cross with S288C, no segregants are expected to grow at the restrictive temperature due to temperature sensitivity of the TS allele. However, in cases where the wild yeast strain harbours variants that can suppress the TS phenotype, the haploid segregants that carry them will be able to grow at the restrictive temperature and will take over the population (Fig 1B).

We first measured the growth defect of each TS allele in complex pools of wild yeast strain cross progeny (Dataset EV1, Appendix Fig

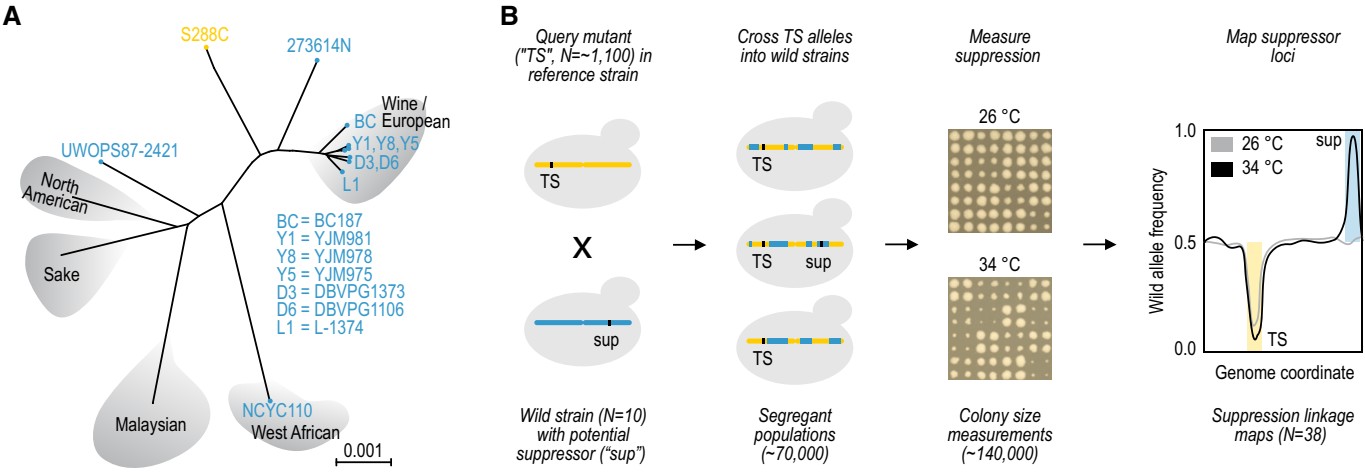

**Figure 1. Experimental overview.**

High-throughput measurement and mapping of suppressor effects from a panel of wild strains.

A  Strains used in the study. Seven wine / European strains, and three other distant ones (all blue) were crossed to the S288C reference (yellow).

B  Strategy for identifying suppression by standing variation. A temperature sensitive (TS) allele collection of ~ 1,100 partial loss-of-function mutants in the reference background (yellow) was crossed to the 10 wild yeast strains with potential suppressor alleles (blue), to produce large segregant populations selected to carry the TS allele. Each cross was performed in one or two biological and four technical replicates. The fitness of the resulting ~ 70,000 populations was measured at 26°C (permissive temperature) and 34°C (restrictive temperature). A subset of 38 candidate suppression events were used in bulk segregant analysis for linkage mapping of causal loci that display selection at restrictive temperature (black) but not permissive temperature (grey).

Data information: Panel (A) adapted from Liti *et al* (2009).

S1). We estimated suppression as normalised $\log_2$-scale growth difference between the wild and reference strain crosses at the restrictive temperature and considered a TS allele phenotype suppressed, if this value was above 0.75, i.e. the wild strain segregants had a 1.68-fold improvement in growth, and if this was unlikely due to chance (Bonferroni-corrected $P = 0.04$, Methods, Dataset EV2).

Our screen included several positive control crosses that were expected to show suppression. Three of the wild strains harbour a chrVIII-chrXVI reciprocal translocation (Pérez-Ortín *et al*, 2002; Fig EV1A). Crossing these strains to the reference strain results in 25% of the progeny carrying a duplication of a substantial part of the left arm of chromosome XVI (Fig EV1B). This creates an extra copy of the essential query genes in this region that complements the TS allele. Reassuringly, we confirmed that this extra copy suppressed 35 out of 53 TS alleles in the duplicated region on average, while segregant progeny from the other wild strains suppressed a median of two (Fig 2A). Further, the extra copy of chromosome VIII carried by the NCYC110 strain resulted in a similar pattern of suppression (Fig EV2A–C).

Also beyond these large genomic determinants, suppression of fitness defects by standing variation in the species was relatively common. Excluding the copy-number suppression events discussed above, 192 of 1,106 TS alleles (17%) and 149 of the 580 tested genes (26%) were suppressed in segregant progeny from at least one genetic background, and on average progeny from a wild strain cross could suppress 37 essential gene mutant alleles (3%). Due to variation in temperature sensitivity, different TS alleles of the same gene are not necessarily expected to show suppression at the same temperature. Nevertheless, if a TS allele was suppressed in segregant progeny from a wild strain, the strongest suppression of another TS allele of the same gene in segregants of the same wild strain was on average substantially higher than expected by chance (0.61 vs 0.39, Appendix Fig S2).

To further validate the suppression events identified in our screen, we tested a selection of 102 suppression effects of variable strength by examining the fitness of hundreds of single colony progeny from individual crosses. As the progeny of most crosses showed high variation in colony size, which was also influenced by the number of colonies on the plate, we used stringent thresholds for identifying suppression (Methods). We observed good concordance of strong effects between the phenotypes of the population in the initial screen and the individual progeny in this assay (62% of crosses with suppression scores above 0.75 in the screen show suppression in the individual colony assay, 16% for crosses with a suppression score below 0.75, Fig EV2E, Dataset EV3).

Thus, we found that suppression by standing genetic variation is relatively common and that the identified suppression events can often be validated by additional TS alleles or in complementary assays.

## Patterns of suppression

Next, we asked whether segregant progeny from genetically similar wild strains were more likely to suppress the same TS alleles compared to more diverse wild strains. Indeed, suppression patterns were more distinct for the two wild strains genetically furthest from the wine/European cluster (maximum Pearson's $R$ to any other wild strain for NCYC110, UWOPS87-2421 less than 0.52, and between

0.64 and 0.77 for the rest) and were consistent with the genetic relatedness otherwise (Fig 2B). The DBVPG1106 wine strain was a phenotypic outlier of Wine/European strains due to overall poor growth at the restrictive temperature (0% of TS allele crosses with $\log_2$-scale colony size of more than 10.5 across all crosses; at least 11% for all other strains, Appendix Fig S1). When using different wild strains as the phenotypic reference to call suppression, the total number of suppressed TS alleles was lower compared to S288C, but their relative frequency across strains was similar (Appendix Fig S3). This suppression pattern is consistent with the reference strain acquiring additional loss-of-function mutations in the laboratory that can be suppressed by crossing to the wild strains, as well as the presence of additional suppression events that are linked to distances in both genetic and phenotypic trees.

The patterns of suppression of the same TS allele or gene in the various wild strain crosses were diverse. In 31% of the cases, the TS allele is possibly temperature-sensitive only in the reference background, with suppression in segregant progeny of most of the tested wild strains (61 of 192 TS alleles with suppression at a lower 0.5 threshold, e.g. TS alleles of *GAB1*, Fig 2C and D). For other TS alleles, suppression was generally limited to a single wild strain (e.g. TS alleles of *NSE4*, Fig 2D). Suppression was also shared across genes with related function. For example, Gab1 is a member of the GPI-anchor trans-amidase complex, mutations to three screened members of which were strongly suppressed (Fig 2E), and the *GPI8* gene with less suppression was likely carrying the suppressor variant (see below). Again, we also observed suppression in specific backgrounds, e.g. mutations to genes in the nuclear condensin complex were suppressed almost exclusively in the UWOPS87-2421 background (Fig 2E). In general, we observed consistent suppression patterns between genes encoding members of the same protein complex most frequently (18% of protein complexes with average between-gene suppression correlation higher than permuted controls, Methods), followed by KEGG pathways (15%) broad functional categories (10%), cellular locations (10%) and Gene Ontology categories (3%). This concordance is consistent with the nature of connectedness within genetic networks in general, where many interactions are shared within complexes, compared to broader functional connections. Finally, we tested whether loss-of-function mutations have accumulated in essential genes in the wild strains that suppress TS alleles of the gene in S288C, but found limited signal (0.14 deleterious mutations in suppressed genes; 0.16 in others).

## Mapping of genomic regions involved in suppression

Given frequent, strong and technically and biologically consistent suppression of TS alleles by variants from wild genetic backgrounds, we next sought to identify the causal loci and genes. First, to estimate the average number of modifiers involved in the suppression phenotype, we dissected meiotic progeny of 16 crosses and examined the growth of spores carrying the TS allele at 26 and 34°C. In all cases, 15–65% of the spores grew well at the restrictive temperature, with little additional phenotypic variation in growth beyond survival, suggesting that most of the detected suppression phenotypes are the result of at most 1–3 strong modifier variants in the wild strain background (Fig EV3, Discussion).

To map the suppressor loci, we performed bulk segregant analysis on 38 segregating TS allele populations at both 26°C

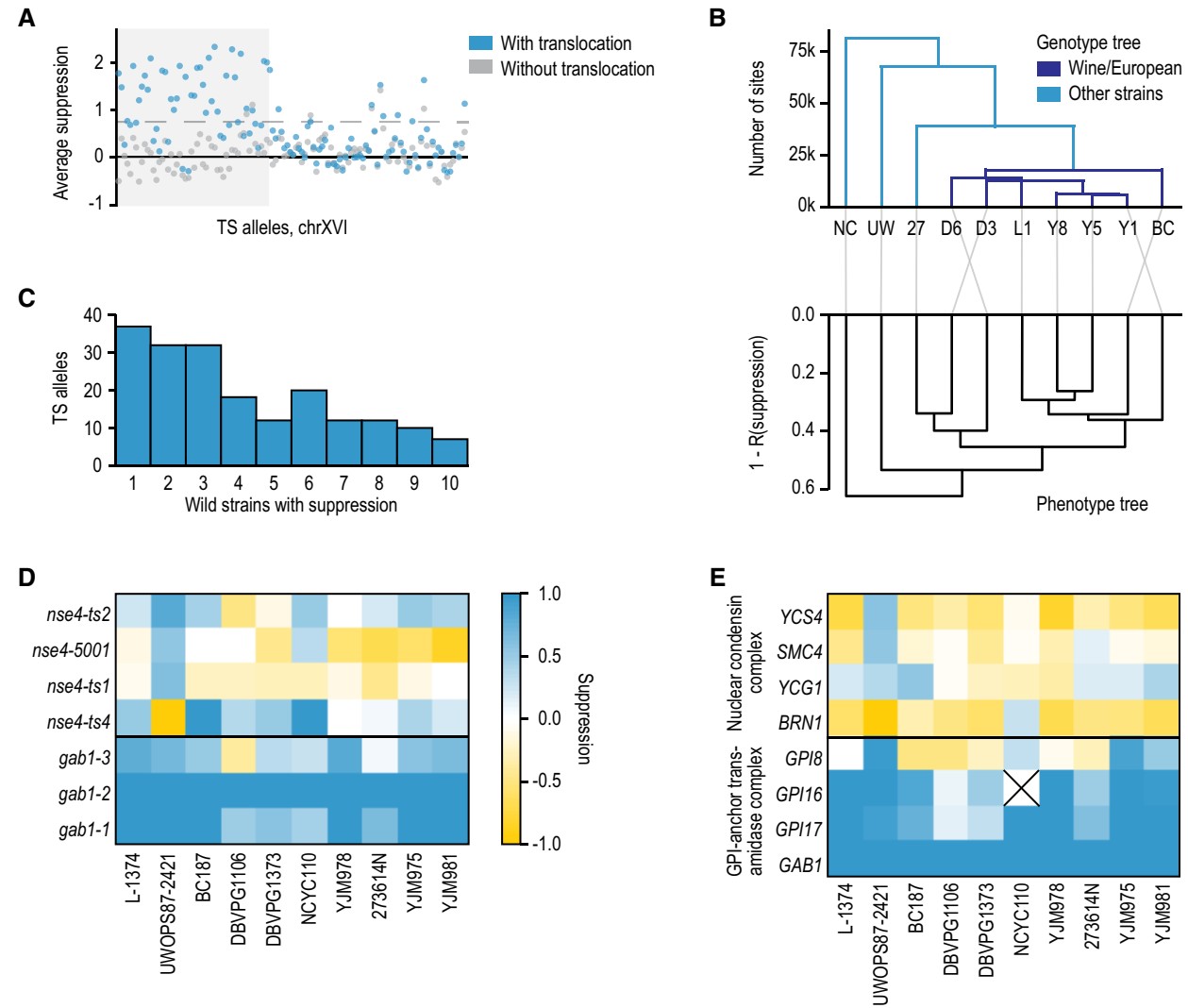

**Figure 2. The extent of genetic suppression of essential gene mutants by standing variation.**

Genetic suppression of essential gene TS mutants is not frequent for individual allele-strain combinations, but relatively common for a gene.

A   A positive control region shows suppression signal. Average suppression score (y-axis) for TS alleles of query genes located on the left arm of chromosome XVI (TS allele index, sorted by chromosome coordinates, x-axis) across strains with a translocation that generates a duplication of the shaded area (blue markers), and strains without the translocation (grey markers). Dashed line: y = 0.75 (suppression cut-off in screen).

B   Genotype and phenotype trees are concordant. Top: Hierarchical clustering (UPGMA) of the ten wild strains used in this study based on sharing segregating sites. Colours: global genetic cluster membership. Bottom: as top, but based on correlation distance between genetic suppression profiles. Strain abbreviations as in Fig 1A, with in addition: NC = NCYC110, UW = UWOPS87-2421, and 27 = 273614N.

C   About a third of suppressed alleles can be explained by a deficiency in the reference strain that is not present in the majority of wild strains. The frequency (y-axis) of the number of wild strains that suppress a TS allele (x-axis) for the 187 alleles that could be suppressed by at least one strain, using a more lenient criterion (suppression > 0.5; no z-score cut-off) to avoid edge effects and winner's curse.

D   Genetic suppression is consistent across different TS alleles of the same gene. Suppression score (colour scale) in crosses to different wild strains (x-axis) for TS alleles (y-axis) of GAB1 and NSE4 genes.

E   Genetic suppression is consistent across genes encoding members of the same complex. Strongest suppression score across TS alleles for a gene (colour scale, as in (D)) in crosses to different wild strains (x-axis) for genes (y-axis) that encode members of the GPI-anchor trans-amidase complex (bottom) or the nuclear condensin complex (top). The GPI16 gene suppression was not estimated in the NCYC110 strain due to chromosome II copy-number variation ("X").

(TS allele functional) and 34°C (TS allele loss-of-function, Fig 1B) (Liti & Louis, 2012). We sequenced the populations and compared variant allele frequencies between the two temperatures (Datasets EV4 and EV5). We first considered positive controls expected to involve suppression by an additional, wild-type copy of the query gene described above, either generated

by the chrVIII-chrXVI translocation (six samples) or located on an aneuploid chromosome (six samples). In all 12 cases, we could indeed observe selection for either the translocation or the aneuploidy and further confirmed that suppression occurred by the presence of a second, wild-type allele of the query gene (Figs EV1C and D, and EV2D).

Second, we sequenced meiotic progeny of nine crosses that showed weak "suppression" in our screen (suppression score below 0.7), unrelated to any known translocations or aneuploidies. Five cases showed selection for newly acquired aneuploidies of either the chromosome carrying the query gene or other loci selected in the crossing protocol. These cases often represent ways of cells to escape the strong selection applied in our protocol, rather than true cases of suppression. The remaining four cases harboured regions of selection for the wild strain sequence specific to high temperature (Datasets EV4 and EV5), suggesting that some of the weaker scores in our screen also represent true cases of suppression and corroborating the observations from the confirmation of individual suppression effects.

Third, we analysed 17 crosses that showed strong suppression in our screen (suppression score above 0.75). The large majority (14) showed regions of specific selection for the wild sequence at high temperature (Fig 3A, Datasets EV4 and EV5), whereas two populations diploidised or showed selection for an aneuploidy. The one remaining cross did not show any suppressor loci or aneuploidies.

Thus, we could map suppressor loci for 14/17 (82%) of the crosses that showed strong suppression in our screen, and in 4/9 (44%) of the crosses that showed weak suppression.

The landscape of suppressors is diverse. We identified 31 suppressor loci in the 19 crosses without aneuploidies (1.6 on average, Figs 3B and 4A, Dataset EV5) and an additional 48 weaker reproducible signals (2.5 on average, Figs 3B and 4A, Dataset EV5). This number of modifier loci is in agreement with our estimates based on segregation patterns observed after tetrad dissection (Fig EV3). Most of the suppressor loci (27/31) were selected for the wild strain sequence, consistent with the additional variation in the species providing the substrate for circumventing essential gene function. Reassuringly, suppressor loci were reproducible across biological replicates, different TS alleles of the same gene when crossed to the same wild strain (e.g. *RPN11*), and same TS alleles when crossed to different wild strains (*TFG1* and *GAB1*; Figs 3C and D, and 4A). A suppressor locus on chromosome XIV was shared across five different essential genes (*GPI13*, *MED7*, *RPN11*, *SEC24* and *TFG1*), indicating the presence of a pleiotropic modifier in this

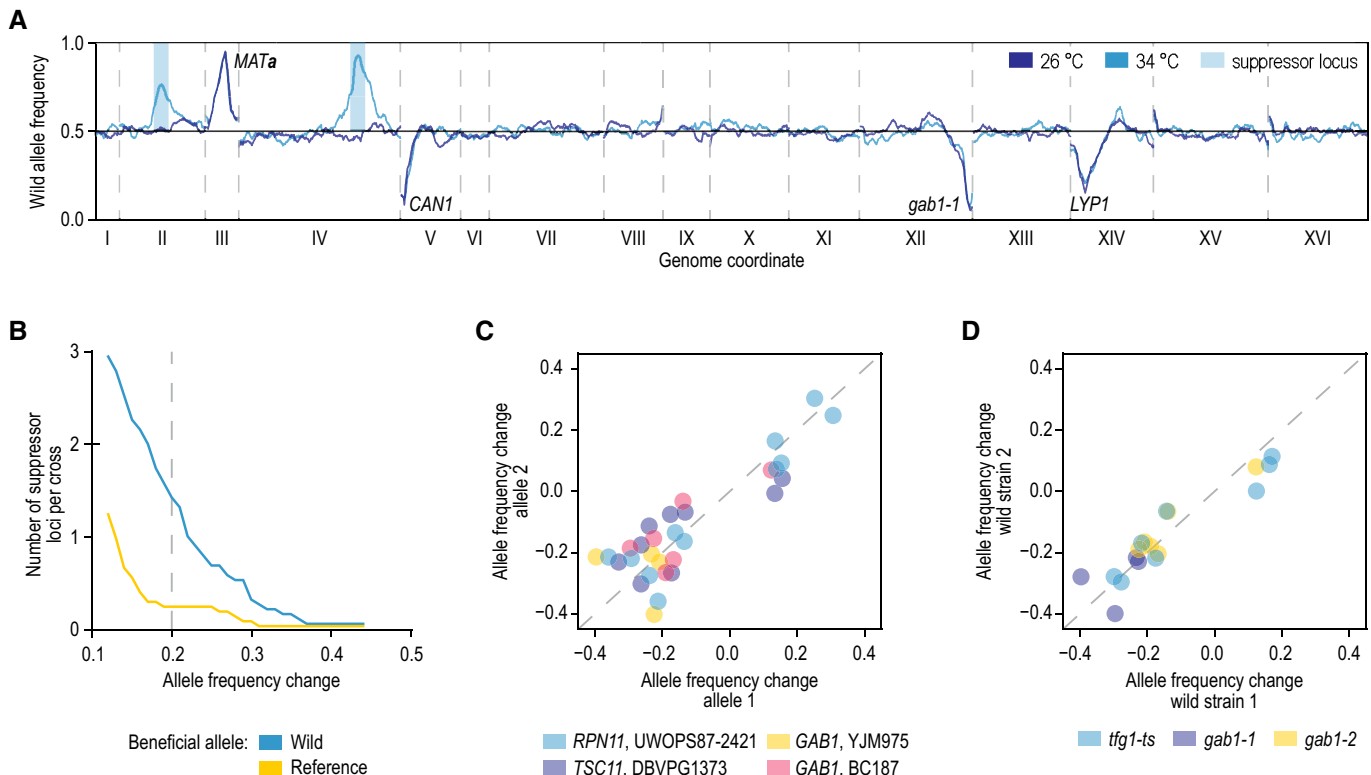

**Figure 3. Mapping suppressor loci by sequencing segregant pools.**

A Example of the mapping results. Wild allele frequency in progeny of a cross of the *gab1-1* temperature-sensitive allele to the YJM975 strain (*y*-axis) along the yeast genome (*x*-axis) at the permissive 26°C (blue) and restrictive 34°C (TS allele loss-of-function, cyan). The allele frequency change between the two temperatures is used in mapping. Labels: selected loci in the cross. Blue regions: called suppressor loci.

B Suppressors are plentiful. The average number of suppressor loci per cross (*y*-axis) at a given allele frequency change cut-off (*x*-axis) with either the wild allele beneficial (blue) or the reference allele beneficial (yellow). Vertical line at an allele frequency change of 0.2: the cut-off used for calling suppressor loci.

C Suppressors are reproducible across TS alleles. Allele frequency change of suppressor loci in crosses using different TS alleles of the same gene (*x*- and *y*-axis) crossed with the same wild strain. Colours: gene and wild strain combinations.

D Suppressors are reproducible across wild strains. Allele frequency change of suppressor loci at 34°C in crosses using the same TS allele and a different wild strain (*x*- and *y*-axis). Colours: TS alleles.

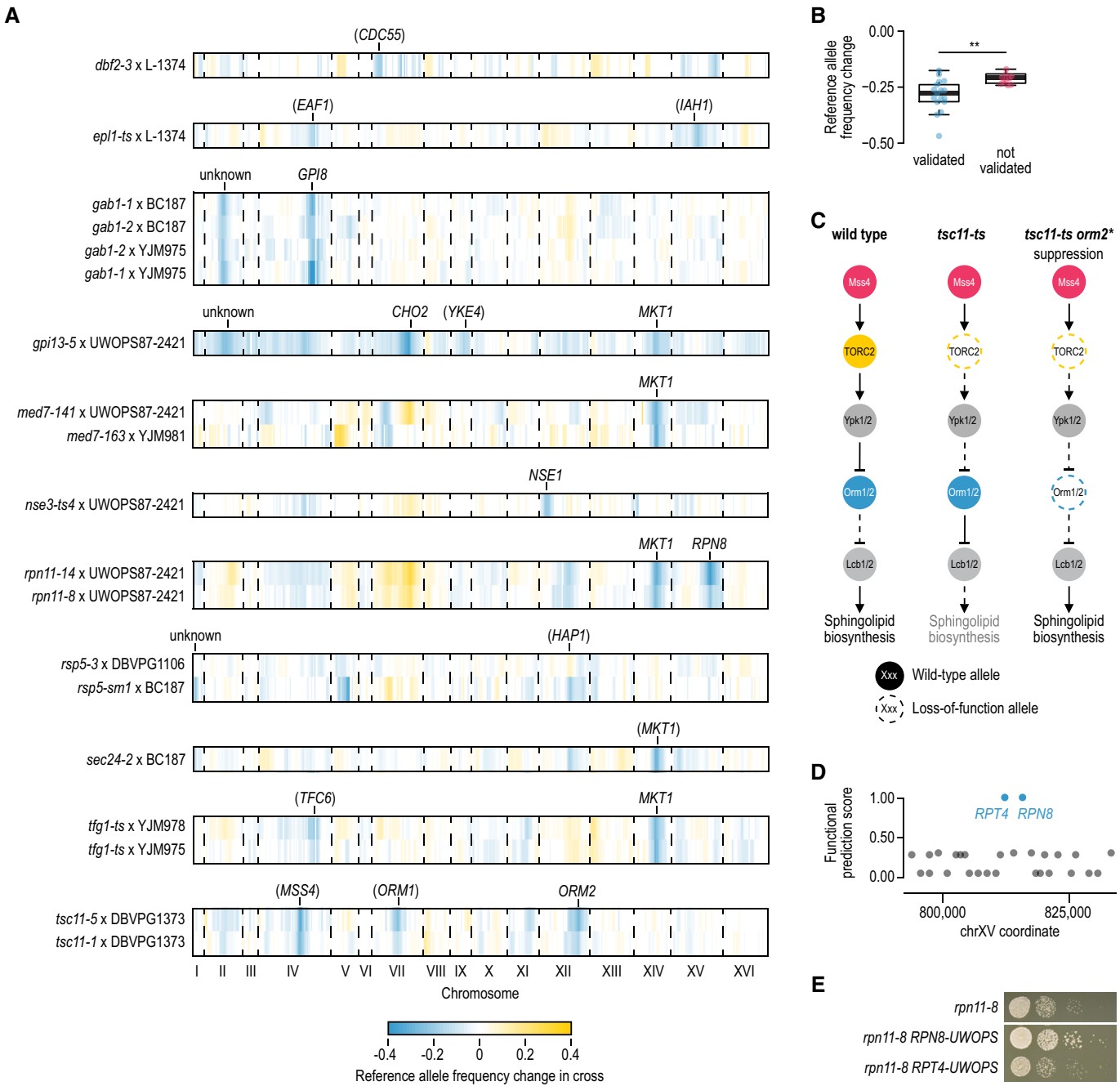

**Figure 4. Identifying and validating suppressor candidates.**

A   Mapping results for segregant pools involving the indicated TS alleles and wild strains. The change in S288C allele frequency between the meiotic progeny isolated at 26°C and 34°C is plotted by genomic coordinate. Causal suppressor genes are indicated for regions that show selection for the wild strain sequence. Genes in brackets have not been validated experimentally.

B   Comparison of the change in reference allele frequency, either for suppressor loci for which a causal suppressor gene was validated ($N = 17$), or for loci for which we were unable to validate a suppressor gene ($N = 10$). Loci for which all experiments failed due to technical reasons were excluded from the analysis. Statistical significance was determined using a two-tailed Mann–Whitney's U-test (**$P < 0.005$). Boxes: first, second, and third quartiles; whiskers: 1.5 interquartile range away from the first and third quartiles.

C   A cartoon of the TORC2 signalling pathway, highlighting suppression of a TORC2 mutant (*tsc11*) by mutations in *ORM2*.

D   Suppressor prediction within the chromosome XV QTL of *rpn11* TS mutants. The functional information prioritisation score (y-axis) for genes in the suppressor region (x-axis) identified *RPT4* and *RPN8* as the two highest-scoring suppressor candidates.

E   Experimental validation of *RPN8* as the causal suppressor of the *rpn11-8* TS mutant. Cultures of the indicated strains were diluted to an optical density at 600 nm of 0.1 and a series of tenfold dilutions was spotted on agar plates and incubated for 2–3 days at 34°C. UWOPS = UWOPS87-2421. See also Fig EV4.

locus, co-localising with the previously characterised *MKT1* gene (Steinmetz *et al*, 2002).

## Suppressor gene identification and validation

We next sought to identify the causal suppressor genes. As each of the mapped regions harbours tens of plausible candidates, we computationally prioritised them based on their functional connection to the query gene (Dataset EV6, Methods, van Leeuwen *et al*, 2020). We also included known general modifiers, such as *MKT1* and *HAP1*, that each affects the expression of thousands of genes (Fay, 2013; Albert *et al*, 2014, 2018; Parts *et al*, 2014). To test the phenotypic consequence of the candidate genes, we replaced their open reading frame and ~ 100–400 bp surrounding region with the wild version in the reference strain background and tested for suppression of the corresponding TS allele. In total, we tested 50 suppressor gene candidates from 31 mapped loci of various strength (Dataset EV7, Fig EV4) and identified causal genes for 17 loci (55%, Dataset EV7, Figs 4A and EV4), validating both our mapping strategy, as well as the computational prioritisation. The 14 suppressor loci without a confirmed suppressor gene resulted from failed experiments, inconclusive results, or cases in which no suppression was observed for the wild allele (Dataset EV7). Causal suppressor genes were more likely to be identified for strong suppressor loci compared to weaker signals, suggesting that in unconfirmed cases the suppression phenotype may have been too weak to detect in our validation assay (Fig 4B).

Many of the validated suppressor genes were consistent with their known roles in the biology of the query gene. For example, two different TS alleles of *TSC11*, which encodes a subunit of TOR complex 2 (TORC2) that activates a phosphorylation cascade controlling sphingolipid biosynthesis, were suppressed by multiple variants present in the DBVPG1373 strain (Fig 4A). The two strongest suppressor loci were located around *MSS4* and *ORM2,* which encode an upstream activator and a member of the TORC2 signalling pathway, respectively (Han *et al*, 2010; Lucena *et al*, 2018; Fig 4A and C). Indeed, we confirmed the suppression of *tsc11-5* temperature sensitivity by the *ORM2-DBVPG1373* allele (Fig EV4). Orm2 inhibits the first committed step in sphingolipid synthesis, and loss of Orm2 function may lead to the reactivation of sphingolipid biosynthesis in the absence of TORC2 (Fig 4C). Intriguingly, a third weak suppressor locus was identified for the *tsc11-5* allele around *ORM1,* a paralog of *ORM2*. However, *ORM1* has no variants within the ORF in the DBVPG1373 strain, and we were not able to confirm a suppression phenotype for the *ORM1* promoter variants in the presence of reference alleles of *MSS4* and *ORM2*.

Genetic mapping alone is not sufficient to identify causal genes. Our computational prioritisation identified *RPT4* and *RPN8* as potential suppressor genes within the chromosome XV suppressor locus of *RPN11* with equally high scores (Fig 4A and D). As the query gene *RPN11* encodes a metalloprotease subunit of the 19S regulatory particle of the proteasome, and the candidate suppressors *RPT4* and *RPN8* also both encode subunits of the same particle, genetic information and computational prior were not sufficient to pinpoint one as the causal suppressor gene. In experimental validations, the *RPN8* allele from UWOPS87-2421 suppressed the *rpn11-8* phenotype, whereas the *RPT4* allele did not (Figs 4E and EV4). Rpn8 and Rpn11 form an obligate heterodimer (Bard *et al*, 2018), and the

*RPN8* allele from UWOPS87-2421 may thus restore the interaction between the two proteins, which could have been weakened by the *RPN11* mutations. This ability to resolve a causal gene from multiple linked candidates underscores the importance of thorough experimental validation to understand the mechanism of suppression.

## Genetic simplicity of strong suppression

Previous approaches for identifying suppressors have relied on spontaneous mutation, and thus sample genetic backgrounds that are very similar to that of the reference. As a result, more complex allele arrangements that may be required for suppression, e.g. combinations of two or more mutations, are not easily obtained. Despite observing multiple loci that are involved in the suppression phenotype in each of the sequenced populations (Fig 4A), we found no evidence for the interdependence of one suppressor locus genotype on the presence of another, and all strong suppressors acted in isolation (Figs 4B and EV4; Discussion). For example, both the *RPN8* and the *MKT1* allele from UWOPS87-2421 could individually suppress the *rpn11-14* TS allele to near wild-type fitness (Fig EV4). We did not observe examples consistent with strong suppression by many small effect variants. Conversely, multiple mutations within a locus could be required for suppression. The *ORM2-DBVPG1373* allele that independently suppressed *tsc11-5* carries two missense mutations, P26T and G134S, that affect conserved residues, are predicted to be highly deleterious and are both required for a robust suppression phenotype (Fig EV4).

Next, we compared the suppressor genes identified by our mapping results to those previously found in the reference strain background (Oughtred *et al*, 2019). In cases where we confirmed a candidate suppressor, we also often found prior evidence of suppression in that gene (4/9 unique suppressor-query pairs; Dataset EV7). In all four cases, the suppressor and query gene pairs encoded members of the same protein complex or pathway, and in three cases the suppressor and query proteins interact physically (Dataset EV7). The five suppressor-query gene pairs that had not been previously described included four cases of suppression by the general modifier *MKT1*. We have previously observed a similar prevalence of general suppressor genes that affect the expression of the query mutant among spontaneous suppressor mutations of TS alleles isolated in the reference strain (~ 50% of all suppressor genes; van Leeuwen *et al*, 2016, 2017). Out of the nine suppressor-query pairs, 7 (78%) appear to involve a gain-of-function suppressor allele (Methods; Dataset EV7). When we exclude *MKT1*, three out of 5 (60%) suppressor genes were classified as gain-of-function alleles compared to the reference allele. This relatively large proportion of gain-of-function alleles is consistent with the idea that loss-of-function alleles may be under stronger negative selection in natural populations.

Overall, mechanisms of suppression identified in a laboratory setting mimic those driven by natural variation and can involve identical suppressor genes when considering suppressors that function within the same functional module as the query gene. In addition, despite the presence of multiple selected suppressor regions in nearly every cross, strong suppressor mutations always acted independently of the genetic background. Combined, these observations are consistent with a model where single genes evolve along a lineage, perhaps adapting to the rest of the environmental and

genetic context via multiple gain-of-function mutations, which then in turn gives the derived allele the ability to independently suppress fitness defects of other alleles.

## Mutations in NSE1 can suppress SMC5/6 complex dysfunction

One of our mapped suppressor interactions involved the suppression of a *nse3-ts4* TS allele by the *NSE1* allele from UWOPS87-2421 ("*NSE1-UW*"; Figs 4A and 5A). Nse1, Nse3 and Nse4 form a subcomplex within the highly conserved SMC5/6 complex, which is essential for the removal of recombination intermediates during DNA replication and repair (Fig 5C) (De Piccoli *et al*, 2006; Menolfi *et al*, 2015). Nse3 and Nse4 bridge the globular head domains of Smc5 and Smc6 (Fig 5C), whereas Nse1 is a RING finger protein with ubiquitin ligase activity that strengthens the interactions between Nse3 and Nse4 (Pebernard *et al*, 2008; Hudson *et al*, 2011). The *NSE1-UW* allele also suppressed the growth defect of a *nse4-ts4*

TS allele, but not that of any of the other tested SMC5/6 subunits (Fig 5A). A *nse1* loss-of-function allele exacerbates the fitness defect of a *nse3* TS mutant (Costanzo *et al*, 2016), and overexpression of *NSE1* suppresses a *nse3* TS allele (Magtanong *et al*, 2011) (Fig 5D), suggesting that the *NSE1-UW* allele has a gain-of-function effect that improves the stability or activity of the SMC5/6 complex. Indeed, the *NSE1-UW* allele suppressed the sensitivity of *nse3* and *nse4* mutants to DNA damaging ageing agents hydroxyurea (HU) and methyl methanesulfonate (MMS) (Figs 5B and EV5A), but could not suppress the lethality associated with deleting either *NSE3* or *NSE4* (Fig EV5B). Thus, phenotypic defects of *nse3* and *nse4* partial loss-of-function mutants could be restored by the presence of *NSE1-UW*.

To more directly test the impact of *NSE1-UW* allele on the SMC5/6 complex function, we measured its accumulation at two established chromosomal SMC5/6-binding sites using an Smc6-FLAG-based ChIP-qPCR assay in the reference strain (Lindroos *et al*, 2006; Jeppsson *et al*, 2014). Although the amount of Smc6-FLAG protein

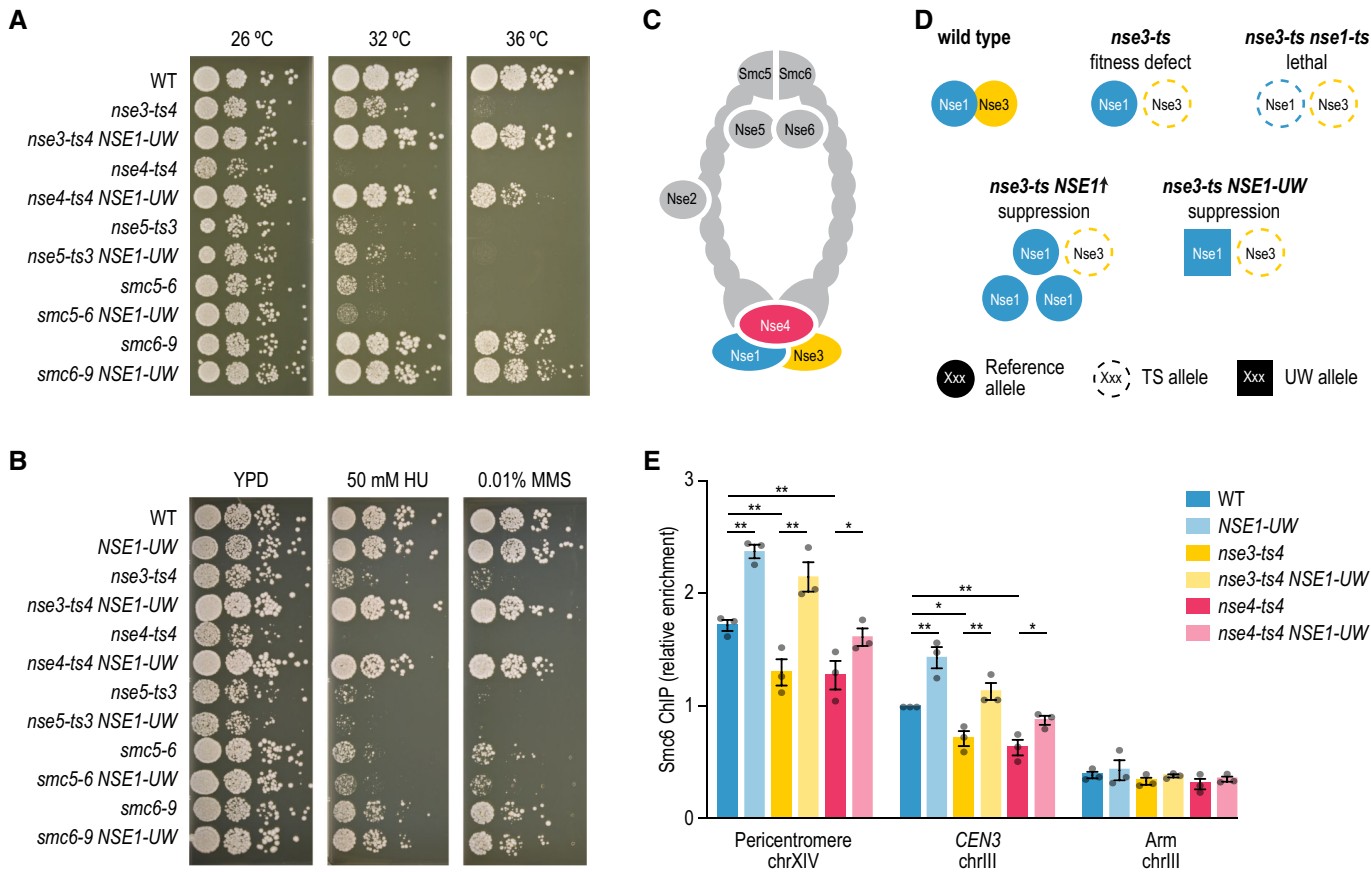

**Figure 5. The *NSE1* allele of UWOPS87-2421 can suppress *NSE3* and *NSE4* TS mutants.**

A, B  Suppression of *nse3-ts4* and *nse4-ts4* temperature sensitivity (A) and DNA damage sensitivity (B) by the *NSE1* allele of UWOPS87-2421. Cultures of the indicated strains were diluted to an optical density at 600 nm of 0.1 and a series of tenfold dilutions was spotted on agar plates and incubated for 2–3 days. UW = UWOPS87-2421. The plates shown in (B) were incubated at 30°C.

C  A cartoon of the SMC5/6 complex.

D  An illustration of the various types of genetic interactions that have been observed between different alleles of *NSE1* and *NSE3*.

E  Recruitment of Smc6-FLAG by ChIP-qPCR at two known SMC5/6-binding sites (pericentromere of chromosome XIV and *CEN3*) and one negative control locus (arm of chromosome III) in G2/M arrested strains. Relative enrichment corresponds to the ratio of the signal after immunoprecipitation (FLAG) over beads alone, normalised to the ratio in wild-type cells at *CEN3*. Error bars: standard error of the mean of three independent experiments. Statistical significance was determined using an unpaired, two-tailed *t*-test (*$P < 0.05$, **$P < 0.005$).

was similar in all strains (Fig EV5C), the accumulation of Smc6-FLAG at the two genomic loci was substantially reduced in *nse3-ts4* and *nse4-ts4* mutants compared to wild type (Fig 5E). Replacing the reference *NSE1* allele with the *NSE1* allele of UWOPS87-2421 increased recruitment of the SMC5/6 complex to the DNA, both in the presence of wild-type or TS alleles of *NSE3* or *NSE4*. This suggests that the *NSE1-UW* allele increases association of the SMC5/6 complex to the DNA, thereby counteracting the negative effects of the *nse3* and *nse4* TS alleles on SMC5/6 complex activity.

# Discussion

We used systematic large-scale genetics to cross partial loss-of-function alleles to ten different genetic backgrounds, measured the extent to which standing variation in the species can suppress the loss-of-function phenotype, used a powerful pooled mapping approach to localise modifying alleles and identified the causal genes for many of them. We found that suppression was consistent across replicates, TS alleles and within complexes, with independent strong modifier genes often acting either directly by interacting with the mutated protein or the complex in which it operates, complementing the output of the pathway in which it is a member, or unspecifically via general compensation mechanisms.

### Genetic architecture of suppression

The genetic architecture of suppression we mapped was skewed towards alleles of strong effect. Nearly, all linkage maps had a strong modifier, and there was a long tail of weaker effects, many of which validated, as has been observed for virtually all mapped traits in general. Some previous studies identified a relatively large fraction of beneficial reference alleles in yeast linkage maps (e.g. $\frac{1}{3}$ of overall in Parts *et al*, 2011). Although we find a similar fraction of selected reference alleles when including weak modifier loci, we found the reference allele was preferred in only every eighth strong suppressor locus. This is consistent with a few large effect alleles explaining the majority of phenotypic variability, that then by definition requires their effect to align with the phenotype differences between the strains.

We attempted to identify causal genes for all mapped strong suppressor loci and succeeded for most. Multiple TS alleles and different wild strains often consistently supported the suppression region, and the plausible suppression mechanisms were affected via gain-of-function mutations affecting complex integrity, pathway activity, or unspecific modifiers. For the three strong suppressor loci for which we failed to confirm a suppressor candidate (Fig 4A), we may have failed to include the causal suppressor gene in our experiments, or the suppression may have been dependent on the presence of other suppressor alleles. However, cases of suppression that require more than a single allele were not frequent in our analysis, as a single strong suppressor allele could independently overcome the mutation phenotype in nearly all mapped cases. These results are consistent with a long list of studies that identify a single gene or genomic locus with a strong effect on a phenotypic trait in diverse organisms (Johnston *et al*, 2011; Barson *et al*, 2015; Jones *et al*, 2018; Thompson *et al*, 2020) and suggest additional scrutiny of single large effect alleles modifying human phenotypes as a promising research direction as well.

At first glance, the identification of a single strong suppressor locus in most crosses is in contrast with other studies that have described complex background dependence of mutation effects (Mullis *et al*, 2018; Hou *et al*, 2019). However, we believe that these two sets of results are consistent with a view where there is polygenic background dependency, which is sometimes (in our data, not more than 4%, the frequency at which allele–strain combinations show strong suppression) peppered with large effect modifier alleles. The apparent discrepancy likely arises due to our selection of crosses to use for mapping, as we prioritised suppression events where the phenotypic advantage was large, thus also implicitly biasing the selected populations for large genetic effects. While we report the strongest linkages, there are also additional weaker ones that are reproducible (Fig 3). Thus, our maps confirm the complexity of background dependence observed before in the large number of mapped loci, but also present a simplicity in the small number of loci that explain substantial variation and can act independently.

Many of the suppression events were driven by aneuploidies. These generally involved pre-existing aneuploidies and translocations, most frequently in the wild parent. This is not surprising, as the wild strains generally tolerate aneuploidy well (Hose *et al*, 2015; Peter *et al*, 2018), and the strong imposed selection forces the cells to use all available diversity to survive. The range of possible ways to escape the various selection steps was such that it is arguable that most of the logically consistent and physically possible scenarios took place. While such chromosome-scale plasticity may not be common in higher eukaryotes where imbalances in gene dosage are often deleterious, it underscores the evolutionary potential of large-scale rearrangements compared to point mutations. *De novo* aneuploidies and diploidization events also explain a large fraction (56%) of the weak "suppression" signals that we sequenced, with a suppression score < 0.7 in our screen, where a few cells had escaped one of the selection steps, and could partially take over the population. These events could also explain the suppression signals that were not confirmed by individual segregant colony analysis (Dataset EV3, Fig EV2E), where the number of examined segregants was in the low hundreds, so that infrequent escapes plausibly present in a complex pool of millions of cells were unlikely to occur.

### Consistency of suppression across experiments

A subset of the genes we mapped suppressors for had previously been analysed to identify spontaneously evolved modifiers in the reference background. The suppressor genes that had a functional connection to the query gene were often identical in both studies, consistent with the shared selection targets of *de novo* and pre-existing variation observed under drug treatment (Li *et al*, 2019). This could indicate that the suppressor allele complements an independent deficiency in the reference strain (consistent suppression of *GAB1* in all wild strains by the same *GPI8* allele) or that the suppressor has co-evolved with the complex or pathway within (SMC5/6 complex and TORC2 pathway). Further, the vast majority of validated suppressor alleles likely conferred a gain-of-function phenotype compared to the reference allele (Dataset EV7), whereas many of the spontaneous suppressor mutations isolated in the reference background had a loss-of-function effect (van Leeuwen *et al*, 2016, 2020). Loss-of-function alleles are more likely to arise spontaneously as the underlying mutation events are more common, but

may have a higher chance to be subjected to negative selection in natural populations compared to gain-of-function variants. Alternatively, the high fraction of gain-of-function effects among natural suppressors may result from a loss-of-function defect of the gene in the reference background. This is likely true for the *HAP1* and *MKT1* alleles (see below). However, we did not find evidence for this being the dominant mechanism, as two thirds of the suppressed genes showed suppression by natural variation in only a minority of the wild backgrounds (Fig 2C).

We also frequently observed suppression via general, pleiotropic modifiers. Although general modifiers that can suppress the growth defect of many different mutant genes have been identified by spontaneous mutation in the reference background as well, they tend to affect mRNA and protein degradation pathways (van Leeuwen *et al,* 2016, 2017). The natural variation general modifiers *HAP1,* encoding a transcription factor regulating the response to haem and oxygen, and *MKT1,* encoding a nuclease-like protein of which the precise cellular function remains unclear, were never found as spontaneous suppressors among the > 2,000 described suppressor interactions in S288C (van Leeuwen *et al,* 2016). In the case of *HAP1,* this is expected as the gene is inactivated by a transposon insertion in S288C. The *MKT1* gene on the other hand is intact in S288C, but the reference allele may perform poorly compared to *MKT1* alleles available in the wild, that have been described to suppress many different phenotypes, including temperature sensitivity (Steinmetz *et al,* 2002; Fay, 2013; Albert *et al,* 2014, 2018; Parts *et al,* 2014).

Our study used temperature-sensitive mutant strains that show a progressive decline in gene function with an increase in temperature. This enables identifying suppressors that can completely bypass gene function, but also those that rescue partially functional alleles. For example, the *nse3-ts4* allele could be completely rescued by mutations in *NSE1,* both encoding members of the Nse1–Nse3–Nse4 complex module (Fig 5). However, this subcomplex would not assemble in the absence of the *NSE3* gene, and the *NSE1* mutant allele does not rescue a *nse3Δ* deletion mutant (Fig EV5). Comparison of our data with a systematic survey of bypass suppression of essential gene deletion mutants (van Leeuwen *et al,* 2020) showed little overlap in the identified suppressable essential genes, suggesting that the vast majority of standing variation suppressors will depend on the presence of the TS allele.

### Future perspectives

Our screen for natural variants that can suppress TS alleles was not saturated. First, although the TS mutant strain collection we used in our screen contained TS mutants for ~ 60% of all essential yeast genes, due to variation in temperature sensitivity, not all tested genes will have had a suppressable phenotype at our chosen restrictive temperature of 34°C. Second, the set of possible suppressor mutations we considered was restricted to the standing variation in the ten wild strains we used. Indeed, we could not detect all known suppression alleles that have been identified via spontaneous mutation in the reference background. Despite these limitations, we found that 26% of the tested essential genes could be suppressed by at least one wild strain. As this relatively high number is likely an underestimate of the true suppression potential of standing variation, we expect suppression to be common in natural populations.

We have provided a first glimpse into the extent, complexity and mechanisms of mutation effect suppression by standing variation. Given the high frequency at which we observed suppression via complementing natural variants, we expect it to have an important contribution to other phenotypes, species and contexts, including human disease. The large overlap between natural suppressor variants and those identified in a laboratory setting suggests that suppressor screening in human cell lines will help understand variable penetrance of human disease mutations as well. In parallel, systematic studies in yeast and other species will continue to refine our view of the mechanisms adopted by modifier mutations to determine the severity of genetic traits.

# Materials and Methods

### Yeast strains, plasmids and growth

Yeast strains were grown using standard rich (YPD) or minimal (SD) media. Methyl methanesulfonate (MMS) and hydroxyurea (HU) were obtained from Sigma-Aldrich.

For SGA analysis (see below), we used a collection of temperature-sensitive mutants of essential genes (*MATα xxx-ts::natMX4 can1Δ::STE2pr-SpHIS5 lyp1Δ his3Δ1 leu2Δ0 ura3Δ0 met15Δ0*; (Costanzo *et al,* 2016)). Four of these strains appeared to have a different TS mutant allele than originally annotated. Because we could not determine where a potential mistake or mix-up had occurred, we assigned new strain IDs to these strains. TSQ2353 (*tre2-5008*) was renamed as TSQ2884x (*tfg1*), TSQ1864 (*brr2-5019*) as TSQ2885x (*fas2*), TSQ1877 (*iki3-5008*) as TSQ2886x (*epl1*) and TSQ1879 (*iki3-5010*) as TSQ2887x (*epl1*).

For the allele swaps (see section "suppressor candidate validation"), we used strains from either the BY4741 deletion mutant collection (*MATa xxxΔ::kanMX4 his3Δ1 leu2Δ0 ura3Δ0 met15Δ0*; Euroscarf), or the TS-allele-on-plasmid collection (*MATa xxxΔ::natR_kanR(Cterm) his3Δ1 leu2Δ0 ura3Δ0 [xxx-ts_kanR(Nterm), AgMFA2pr-hphR, URA3]*; (van Leeuwen *et al,* 2020)).

All other yeast strains used in this study are listed in Dataset EV8.

### Making the wild yeast strains SGA compatible

Twenty-six wild yeast strains had previously been deleted for *HO* and *URA3*, and haploid *MATa* spores had been isolated (*MATa hoΔ::hphMX6 ura3Δ::kanMX4*; (Cubillos *et al,* 2009)). To make these strains compatible with SGA analysis and facilitate further genetic manipulations, we (partially) deleted the *LEU2* and *HIS3* genes.

First, to delete *LEU2,* we used plasmid p7410 (Dataset EV8) that contains in the following order: a SwaI restriction site, base pair −403 to eight of *LEU2,* base pair +62 to +258 downstream of the *LEU2* stop codon, the *TDH3* promoter from Ashbya gossypii (Ag) driving the *nrsR* ("*natR*") gene followed by the *AgTDH3* terminator, the *GAL1* promoter driving *KAR1* followed by the *AgCYC1* terminator, the *kanMX4* cassette and base pair +62 to +783 downstream of the *LEU2* stop codon. We digested the plasmid using SwaI and transformed the wild yeast strains with the linearised plasmid. Transformants were isolated on YPD + NAT and subsequently replica plated onto YPGal media, to induce overexpression of *KAR1,*

which is lethal and thus selects for recombination between the two *LEU2-3'* sequences. Recombination was confirmed by testing for growth on YPD + G418 media.

Second, to partially delete *HIS3*, we used plasmid p7411 (Dataset EV8), that contains in the following order: a SwaI restriction site, base pair 137 to 310 of the *HIS3* gene, base pair 495 to +112 of the *HIS3* gene, the *AgTEF1* promoter driving *LEU2* followed by its endogenous terminator, the *URA3* gene under control of its own promoter and terminator, and base pair 495 to +707 bp of *HIS3*. We digested the plasmid using SwaI and transformed the wild yeast strains with the linearised plasmid. Transformants were isolated on SD -Ura -Leu and subsequently replica plated onto media containing 5-fluoroorotic acid (SD + 5-FOA), which is toxic to cells expressing *URA3* and will thus select for recombination between the two *HIS3-3'* sequences. Recombination was confirmed by testing for growth on SD -Leu media.

Proper deletion of *LEU2* and a part of *HIS3* was confirmed by PCR. Strain identity was validated by sequencing the barcodes inserted at the *ura3Δ* locus (Cubillos *et al*, 2009). In total, we obtained 10 wild yeast strains with the genotype *MAT**a** ho*Δ*:: hphMX6 ura3Δ::kanMX4 his3Δ1 leu2Δ0* (Dataset EV8).

## Synthetic genetic array (SGA) analysis

Synthetic genetic array analysis was performed as described previously (Baryshnikova *et al*, 2010), with the exception that 5% mannose was added to the YPD plates used in the first steps of SGA analysis to facilitate pinning of the wild isolates. In brief, the 10 SGA-compatible *kanMX*-marked wild strains (Dataset EV8, *MAT**a** ho*Δ*::hphMX6 ura3Δ::kanMX4 his3Δ1 leu2Δ0*), and a S288C negative control strain (DMA1, *MAT**a** his3Δ::kanMX ura3Δ0 leu2Δ0 met15Δ0* or DMA809, *MAT**a** ho*Δ*::kanMX his3Δ0 ura3Δ0 leu2Δ0 met15Δ0*; Dataset EV8) were crossed to a collection of 1,474 *natMX*-marked temperature-sensitive mutants of essential genes (*MATα xxx-ts::natMX4 can1Δ:: STE2pr-SpHIS5 lyp1Δ his3Δ1 leu2Δ0 ura3Δ0 met15Δ0*; (Costanzo *et al*, 2016)). Each cross was performed in four technical replicates, and for six wild strains (Y14273, Y14274, Y14275, Y14276, Y14277 and Y14280; see Dataset EV8 for strain information) and the S288C control, we performed an additional biological replicate, also containing four technical replicates. In a series of subsequent pinning steps, diploid cells were selected and sporulated, and colonies consisting of pools of around 60,000 haploid segregant progeny (Parts *et al*, 2014) carrying both *natMX* and *kanMX* selection markers were isolated. The final selection step for haploid progeny carrying both markers was performed at both 26 and 34°C.

Plate images were processed with gitter v1.0.3 (Wagih & Parts, 2014) and normalised with SGAtools (Wagih *et al*, 2013). Briefly, this process includes processing plate image files to detect the grid of colonies, quantifying the colony sizes, filtering out missing values and any technical replicates that accounted for at least 90% of the variation in the signal, log$_2$-transforming and calculating the average and standard error of the mean of the remaining technical replicates (reported in Dataset EV1). For the six out of ten wild strains that had two biological replicates, we fit the second replicate onto it using linear regression to correct for batch effects. When only a single replicate was present, we reported the mean and standard error of the mean of technical replicates. To obtain a posterior estimate of the fitness effect when two biological replicates were present, we combined the variances estimated from technical replicates of each biological replicate, and the between-replicate difference, treating technical and biological noise as independent, as follows. Modelling the posterior replicate 1 mean estimate as N(m1, v1), where N is the Gaussian distribution, m1 is the mean of technical replicates, and v1 is the squared standard error of the mean; and replicate 2 mean estimate as N(m2, v2), we first estimated biological noise as $v = (m1-m2)^2/2$ as the standard variance estimate. This leads to per-replicate posterior mean estimates of N(m1, v1 + v) and N(m2, v2 + v). We then combine these posteriors. Denoting p1 = 1/(v1 + v), p2 = 1/(v2 + v), w1 = p1/(p1 + p2), and w2 = p2/ (p1 + p2), we report the mean and standard deviation of the fitness of the strain taken from N(w1*m1 + w2*m2, 1/(p1 + p2)). This corresponds to a weighted average of biological replicates, where the less noisy replicate is trusted more.

We filtered out 76 strains that were either missing in the reference cross or all other samples at the restrictive temperature after filtering. We further filtered out 292 temperature-insensitive query strains that did not show lower fitness at the restrictive temperature in the reference strain background (fitness difference between relative colony sizes at 26 and 34°C below 0.2), retaining 1,106 query strains in total for 580 genes.

## SGA suppression analysis

To estimate suppression of the mutation effect by a wild strain, we quantified the difference in fitness (normalised log$_2$-scale colony size) at the restrictive temperature after adjusting for overall growth differences between the reference and wild strains. To adjust for global growth differences, we set the median restrictive temperature fitnesses of temperature-insensitive strains (see previous section) to be equal and scaled the wild strain restrictive temperature fitnesses to minimise mean-squared error of the fit to the respective values of reference. To obtain posterior variance estimates of suppression, we used the posterior normal distributions of the fitnesses of the wild and reference strain crosses to obtain the posterior distribution for the difference, treating them as independent (thus adding variances), and computed its mean and standard deviation. We also calculated *z*-scores of suppression (for the mean suppression to be above 0) as mean divided by standard deviation. We called a TS allele suppressed if the mean adjusted fitness in the wild strain cross was at least 0.75 larger at 34°C than the S288C reference (i.e. colonies on average 1.68 times bigger), and the *z*-score was at least 4.5. We decided on a cut-off of 0.75 as this corresponds to a visually clear difference. The z-score of 4.5 corresponds to a nominal one-sided *P*-value of $3.4 \times 10^{-6}$ for a normal distribution, and a Bonferroni-corrected *P*-value of 0.0359 after adjusting for 10,554 total tests corresponding to 10 wild strains crossed to the 1,106 queries, after filtering out missing values.

To generate genotype and phenotype trees, we used the scikit-learn average() function to compute the UPGMA tree, and the dendrogram() function for display (Pedregosa *et al*, 2012). For genotype trees, we calculated the distance between strains as the number of called genetic variants that are present in either strain, but not the other one. For phenotype trees, we calculated the distance between strains as 1 minus the Pearson's correlation of their suppression profiles.

To test for consistency of suppression within complexes and pathways, we considered multiple functional annotation datasets.

The sources for these datasets were: protein complexes (the Complex Portal (Meldal *et al*, 2019), downloaded June 6, 2018), KEGG pathway annotation (Kanehisa *et al*, 2016), co-expression (gene partners with a co-expression score > 1; (Huttenhower *et al*, 2006)) and subcellular localisation (Huh *et al*, 2003). For each annotation that groups multiple genes, we calculated the average correlation of suppression values across wild strains between all possible allele pairs. To evaluate the significance of these values for a complex with $N$ alleles, we sampled the matching number of $N(N–1)/2$ allele pairs 1,000 times, while also matching the number of pairs that came from the same gene. We calculated the *P*-value of enrichment as the frequency of observing statistics from permuted data more extreme than the real value and used the false discovery rate correction to adjust the *P*-values. We called a gene set consistent above chance if the average correlation was above 0.25, and the corrected *P*-value below 0.25.

To assess whether there is additional deleterious mutation load for genes that can be suppressed, we downloaded the SIFT (Ng & Henikoff, 2001) annotations from the Saccharomyces Genome Resequencing Project 2 (Bergström *et al*, 2014) website http://www. moseslab.csb.utoronto.ca/sgrp/download.html. We calculated the average number of mutations classified as DELETERIOUS in the genes of wild strains that exhibited suppression of some TS allele of that gene, and the average of other gene/strain combinations.

### Random sporulation assay

A total of 102 wild strain × TS allele combinations were selected for confirmation assays (Dataset EV3). Between 3 and 20 different TS alleles were tested for each of the 10 wild strains, for a total of 78 different TS alleles corresponding to 56 different essential genes. The selected crosses spanned a wide range of suppression scores and included eight crosses with a negative suppression value. As controls, we crossed each selected TS allele to a reference S288C strain, and each wild strain was crossed to a wild-type S288C reference strain, giving a total of 78 S288C TS allele controls, and 10 S288C × wild strain controls.

All 190 strain pairs were crossed and sporulated. Sporulated cells were plated onto two agar plates that selected for haploid *MAT**a*** spores that carried the TS allele (SD -His/Arg/Lys +CAN/LYP/NAT). One plate was incubated at 26°C and one at 34°C. After 3 days plates were imaged, colony size and number were determined using CellProfiler (Carpenter *et al*, 2006). We calculated the difference between the number of colonies and the colony area at 26°C and 34°C for each TS allele–wild strain combination and compared the values for the S288C control to those of the wild strain crosses (Dataset EV3). Images that contained < 100 colonies at 26°C were excluded from the analysis, and all images with < 30 colonies were excluded from colony size determination. A TS allele–wild strain pair was considered to show suppression when either the number or the average size of the colonies of the wild cross was substantially larger than that of the control cross (TS allele × S288C) at 34°C (Dataset EV3).

### Sequencing, read mapping, SNP calling and QTL analysis

We selected 38 crosses that showed various levels of suppression in the screen for bulk segregant analysis. The 38 samples included six positive controls involving query genes located on the left arm of chromosome XVI that were suppressed by the chrVIII-chrXVI translocation, six positive controls involving genes located on chromosome II or chromosome VIII that were suppressed by one of the NCYC110 aneuploidies, nine cases that showed weak "suppression" in our screen (suppression score < 0.7), and 17 cases that showed strong suppression in our screen (suppression score > 0.75). In addition, we crossed each wild strain to a S288C reference strain. We collected at least two replicates of 1,000 haploid progeny colonies per temperature for each cross, using the random sporulation assay outlined above. Colonies were scraped from the agar plates, and genomic DNA was isolated from the pools using the Qiagen DNeasy Blood & Tissue kit. Samples were sequenced using Illumina sequencing.

For each bulk segregant sequencing sample, we performed read mapping and variant calling under the Varathon framework (https:// github.com/yjx1217/Varathon). Briefly, the raw reads were trimmed by trimmomatic v0.38 (Bolger *et al*, 2014) and subsequently mapped to the yeast reference genome (SGD R64-1-1) using bwa v0.7.17 (Li & Durbin, 2009). The resulting read alignment was further processed by samtools v1.9 (Li *et al*, 2009), picard tools 2.18.25 (https://broadinsti tute.github.io/picard/) and GATK3 v3.6 for sorting, duplicate removal, INDEL realignment and indexing. Variant calling was carried out by freebayes v1.2.0 (Garrison & Marth, 2012) with the customised options "--ploidy 1 --min-alternate-fraction 0 --genotype-qualities". Raw variant calls were processed by vt (github commit version f6d2b5d) (Tan *et al*, 2015) for variant decomposition, normalisation, annotation and filtered by vcffilter (distributed together with free-bayes) with the filter: "QUAL > 30 & QUAL / AO > 1 & SAF > 0 & SAR > 0 & RPR > 1 & RPL > 1". Finally, VEP 101.0 (McLaren *et al*, 2016) was used to evaluate the functional impact of each variant by leveraging its specific genomic context.

We stratified the 38 bulk segregant QTL mapping experiments according to genomic coverage and screen signal. We separated the six crosses with the NCYC110 strain due to the wild strain ploidy issues, seven further crosses that had evidence for aneuploidy from sequencing coverage, and a final six crosses with chrXVI-VIII translocation that creates an additional wild-type copy of the query gene in the segregants. To call QTLs in the remaining 19 samples without ploidy issues, and with strong or moderate suppression scores in the screen, we used Selection QTL Mapper (https:// github.com/PMBio/sqtl), which implements the approach used for bulk segregant analysis mapping described in Parts *et al* (2011). Briefly, this approach first estimates reference allele frequencies in each sample using a probabilistic model that includes allele frequencies as latent variables, sequencing reads as observations and the recombination rate parameter to couple frequencies at nearby sites. The posterior allele frequency distributions were then combined across biological replicates according to Bayes rule and used to identify a broad set of QTL regions that had at least 12% frequency change between permissive and restrictive temperatures, and were at least 1kb long, using parameters "af_lenient=0.8, sd_lenient=3, af_stringent=0.12, sd_stringent=5, length_cutoff=1000, peak_cut-off=0.03". A stricter set with allele frequency change of at least 0.20 was used for all but reproducibility analyses. Sites within 30kb of the TS allele or an SGA selection marker were not considered as QTL candidates.

All whole-genome sequencing data are publicly available at NCBI's Sequence Read Archive (http://www.ncbi.nlm.nih.gov/sra),

under accession number PRJNA673501. Variant frequencies are listed in Dataset EV4.

## Suppressor gene prediction

For each detected QTL, we predicted the potential causal suppressor genes by ranking the genes for which the allele frequency change was within 3% of the strongest selected variant in the region by their functional relationship to the query gene, as described previously (van Leeuwen *et al*, 2020). In addition, we scored essential candidate genes higher than nonessential genes. Briefly, we evaluated the following functional relationships and gene properties in this order of priority: co-complex (highest priority), co-pathway, co-expression, co-localisation and essentiality of the suppressor candidate (lowest priority). Thus, genes with co-complex relationships were ranked above those with only co-pathway relationships. Additionally, the order between genes within a given set was established by evaluating the rest of the functional relationships. For instance, the set of genes that were co-expressed with the query gene, but not in the same complex or pathway, were further ranked by whether they co-localised (highest rank) or not (lowest rank) with the query. The sources for these datasets were protein complexes (the Complex Portal (Meldal *et al*, 2019), downloaded June 6, 2018), KEGG pathway annotation (Kanehisa *et al*, 2016), co-expression (gene partners with a co-expression score > 1; Huttenhower *et al*, 2006) and subcellular localisation (Huh *et al*, 2003). We manually added suppressor candidate genes with genetic interactions or other known functional connections to the query gene that were not captured by our computational prediction and also included known general modifier genes *MKT1* and *HAP1*.

## Suppressor candidate validation

To validate the predicted suppressor genes, we introduced 50 potential suppressor alleles into the reference genetic background. First, *kanR* or *nrsR* ("*natR*") targeting guide RNA (gRNA) sequences were cloned into the pML104 or pML107 plasmid vectors, which carry Cas9 and either *URA3* or *LEU2* (Dataset EV8, Laughery *et al*, 2015). Second, for nonessential suppressor gene candidates, we amplified the genes including ~ 400 bp upstream of the start codon and ~ 400 bp downstream of the stop codon from the various wild strains by PCR and co-transformed the PCR fragment and the pML104-kanR1136 and pML107-kanR468 plasmids (Dataset EV8) into a strain carrying a deletion allele of the suppressor gene (*MAT**a** xxxΔ::kanMX4 his3Δ1 leu2Δ0 ura3Δ0 met15Δ0*; Euroscarf). The gRNAs will cut the *kanMX4* cassette at two places and the homology of the promoter and terminator sequences of the PCR product to the genomic sequences flanking the double-stranded DNA breaks will promote repair via homologous recombination and integration of the PCR product into the genome. For essential genes, we used a similar strategy using a set of haploid strains in which the essential gene of interest was deleted in the genome but present on a plasmid (*MAT**a** xxxΔ::natR_kanR(Cterm) his3Δ1 leu2Δ0 ura3Δ0 + [XXX, URA3]*; van Leeuwen *et al*, 2020), and the plasmids pML104-natR412 and pML107-natR854 (Dataset EV8) that carry gRNAs that target *natR*.

Transformants were initially selected on SD -Ura -Leu and then propagated on YPD. Within 3 days of growth on YPD, the vast majority of yeast strains had lost the gRNA plasmids and properly replaced the suppressor candidate deletion allele with the wild allele, which we confirmed by PCR. For essential genes, we streaked the allele-swapped strains on SDall + 5-FOA to remove the plasmid carrying the essential gene.

Next, we crossed the allele-swapped strains to the corresponding TS mutant and sporulated the resulting diploids. We isolated haploid progeny carrying the TS allele and confirmed the identity of the suppressor allele by Sanger sequencing. Growth of the TS mutants carrying the suppressor candidate allele from a wild strain was monitored at various temperatures to confirm the suppression phenotype.

Gain- or loss-of-function effects were predicted for each validated suppressor gene based on previously described genetic interactions between the query allele and deletion or overexpression alleles of the suppressor gene (Oughtred *et al*, 2019), or based on known phenotypes of the S288C and wild alleles (i.e. *HAP1* and *MKT1*).

## Smc6-FLAG chromatin immunoprecipitation

Smc6-FLAG strains were constructed by PCR gene-targeting (Longtine *et al*, 1998) using primers AGAGACCCTGAGAGACAGAATAATTCCAATTTTTATAATcggatccccgggttaattaa and GACGATTACACAATATTTTGAATAATTACATGAAGAAACAgcgcgttggccgattcatta to amplify the FLAG-tag from pFA6-6xGLY-3xFLAG-HIS3MX6 (Funakoshi & Hochstrasser, 2009). Proper tagging was checked by colony PCR using primers TGCGGTCAAGGATTATTGCG and CGCTGTGAGAGTTGTTGAGG.

Smc6-FLAG expression was confirmed by Western blotting. For each strain, whole-cell extracts were prepared by TCA precipitation using 10 $OD_{600}$-units of cells and analysed by SDS–PAGE. Western blotting was performed using an anti-FLAG antibody (clone M2, Sigma-Aldrich, catalogue number F1804). Ponceau staining was used as a loading control.

Chromatin immunoprecipitation (ChIP) was performed as previously described with slight modifications (Cobb *et al*, 2003). Briefly, cells were grown to $5 \times 10^6$ cells/ml in YPD and arrested in G2/M by incubation with nocodazole (15 µg/ml, Sigma-Aldrich) for 2 h. Samples were fixed with 1% formaldehyde. Cell pellets were resuspended in lysis buffer (50 mM Hepes, pH = 7.5, 140 mM NaCl, 1 mM Na EDTA, 1% Triton X-100, 0.1% sodium deoxycholate) containing protease inhibitors. Extracts were incubated with Dynabeads mouse IgG (Invitrogen, M-280) coated with antibody against FLAG (clone M2, Sigma-Aldrich) for 2 h at 4°C. DNA was purified and enrichment at specific loci was measured using qPCR. Relative enrichment was determined by $2^{-DDCt}$ method (Livak & Schmittgen, 2001; Cobb & van Attikum, 2010). Dynabeads without antibody were used to correct for background. An amplicon 14 kb downstream of ARS607, devoid of Smc6 binding, was used for normalisation (Lindroos *et al*, 2006). Used primers are listed in Dataset EV8.

## Data availability

The datasets (and computer code) produced in this study are available in the following databases:

- Whole-genome sequencing data: NCBI's Sequence Read Archive, accession number PRJNA673501 (https://www.ncbi.nlm.nih.gov/bioproject/PRJNA673501/).

- Raw colony size data: Figshare (http://dx.doi.org/10.6084/m9.f igshare.14170787)
- Computer code: GitHub (https://github.com/lp2/2021-MolSysBio-YeastSuppressors)

**Expanded View** for this article is available online.

## Acknowledgements

We thank M. Costanzo, J. Hou, S. Soyk, G. Tan, M. Taschner, B. Ünlü and A. Vjestica for critical reading of the manuscript, discussions, reagents and technical assistance. We also thank B. Andrews and C. Boone whose laboratories the work was incepted in, and whose vision shaped the direction of this project. LP was supported by Wellcome (206194), IT Centre of Excellence EXCITE (TK148) and a Marie Curie International Outgoing Fellowship (328541). JL was supported by the Swiss National Science Foundation (PCEGP3_181242). GL was supported by the Foundation for Medical Research (EQU202003010413). CP holds a Ramon y Cajal fellowship (RYC-2017-22959).

## Author contributions

Project supervision: LP and JL; Supervision of specific analyses: PA and GL; Experiments: AB, MLo, MWY, MLa, BJSL, EE and JL; Computational and statistical analyses: LP, JXY, CP and JL; Manuscript preparation: LP and JL; Input for manuscript preparation: All authors.

## Conflict of interest

The authors declare that they have no conflict of interest.

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
