## [Review Process File · Molecular Systems Biology]

Natural variants suppress mutations in hundreds of essential genes

Leopold Parts, Amandine Batte, Maykel Lopes, Michael Yuen, Meredith Laver, Bryan-Joseph San Luis, Jia-Xing Yue, Carles Pons, Elise Eray, Patrick Aloy, Gianni Liti, and Jolanda Van Leeuwen
DOI: 10.15252/msb.202010138

Corresponding authors: Leopold Parts (leopold.parts@sanger.ac.uk) , Jolanda Van Leeuwen (jolanda.vanleeuwen@unil.ch)

Review Timeline:

Submission Date:	24th Nov 20
Editorial Decision:	7th Jan 21
Revision Received:	26th Mar 21
Editorial Decision:	16th Apr 21
Revision Received:	22nd Apr 21
Accepted:	23rd Apr 21

Editor: Jingyi Hou

Transaction Report:

7th Jan 2021

Manuscript Number: MSB-2020-10138

Title: Diverse natural variants suppress mutations in hundreds of essential genes

Author: Leopold Parts

Amandine Batte

Maykel Lopes

Michael Yuen

Meredith Laver

Bryan-Joseph San Luis

Jia-Xing Yue

Carles Pons

Elise Eray

Patrick Aloy

Gianni Liti

Jolanda Van Leeuwen

Dear Dr. Parts,

Thank you for submitting your work to Molecular Systems Biology. We have now heard back from the three reviewers who agreed to evaluate your manuscript. As you will see below, the reviewers think the study is interesting. They raise however a series of concerns, which we would ask you to address in a major revision.

I think that the reviewers' recommendations are rather clear and there is no need to reiterate their comments. All issues need to be satisfactorily addressed. As you may already know, our editorial policy allows in principle a single round of major revision, so it is essential to provide responses to the reviewers' comments that are as complete as possible. Please feel free to contact me in case you would like to discuss in further detail any of the issues raised by the reviewers.

On a more editorial level, please do the following:

- Please provide a .docx formatted version of the manuscript text (including legends for main figures, EV figures and tables). Please make sure that the changes are highlighted to be clearly visible.
- Please provide individual production quality figure files as .eps, .tif, .jpg (one file per figure).
- Please provide a .docx formatted letter INCLUDING the reviewers' reports and your detailed point-by-point responses to their comments. As part of the EMBO Press transparent editorial process, the point-by-point response is part of the Review Process File (RPF), which will be published alongside your paper.
- Please note that all corresponding authors are required to supply an ORCID ID for their name upon submission of a revised manuscript.
- We replaced Supplementary Information with Expanded View (EV) Figures and Tables that are collapsible/expandable online (see examples in <http://msb.embopress.org/content/11/6/812>). A

maximum of 5 EV Figures can be typeset. EV Figures should be cited as 'Figure EV1, Figure EV2' etc... in the text and their respective legends should be included in the main text after the legends of regular figures.

Additional Tables/Datasets should be labeled and referred to as Table EV1, Dataset EV1, etc. Legends have to be provided in a separate tab in case of .xls files. Alternatively, the legend can be supplied as a separate text file (README) and zipped together with the Table/Dataset file.

For the figures and tables that you do NOT wish to display as Expanded View figures, they should be bundled together with their legends in a single PDF file called *Appendix*, which should start with a short Table of Content. Each legend should be below the corresponding Figure/Table in the Appendix. Appendix figures and tables should be referred to in the main text as: "Appendix Figure S1, Appendix Figure S2, Appendix Table S1" etc. See detailed instructions regarding expanded view here: <https://www.embopress.org/page/journal/17444292/authorguide#expandedview>.

-Before submitting your revision, primary datasets (and computer code, where appropriate) produced in this study need to be deposited in an appropriate public database (see <https://www.embopress.org/page/journal/17444292/authorguide#dataavailability>).

The accession numbers and database should be listed in a formal "Data Availability " section (placed after Materials & Method) that follows the model below (see also <https://www.embopress.org/page/journal/17444292/authorguide#dataavailability>). Please note that the Data Availability Section is restricted to new primary data that are part of this study.

Data availability

- We would encourage you to include the source data for figure panels that show essential quantitative information. Additional information on source data and instruction on how to label the files are available at < <https://www.embopress.org/page/journal/17444292/authorguide#sourcedata> >.

- All Materials and Methods need to be described in the main text. We would encourage you to use 'Structured Methods', our new Materials and Methods format. According to this format, the Material and Methods section should include a Reagents and Tools Table (listing key reagents, experimental models, software and relevant equipment and including their sources and relevant identifiers) followed by a Methods and Protocols section in which we encourage the authors to describe their methods using a step-by-step protocol format with bullet points, to facilitate the adoption of the methodologies across labs. More information on how to adhere to this format as well as downloadable templates (.doc or .xls) for the Reagents and Tools Table can be found in our

author guidelines: <

<https://www.embopress.org/page/journal/17444292/authorguide#researcharticleguide>>. An example of a Method paper with Structured Methods can be found here: .

- Regarding data quantification:

Please ensure to specify the name of the statistical test used to generate error bars and P values, the number (n) of independent experiments (please specify technical or biological replicates) underlying each data point and the test used to calculate p-values in each figure legend. Discussion of statistical methodology can be reported in the materials and methods section, but figure legends should contain a basic description of n, P and the test applied.

Graphs must include a description of the bars and the error bars (s.d., s.e.m.).

- Please provide a "standfirst text" summarizing the study in one or two sentences (approximately 250 characters, including space), three to four "bullet points" highlighting the main findings and a "synopsis image" (550px width and max 400px height, jpeg format) to highlight the paper on our homepage.

Here are a couple of examples:

<https://www.embopress.org/doi/10.15252/msb.20199356>

<https://www.embopress.org/doi/10.15252/msb.20209475>

<https://www.embopress.org/doi/10.15252/msb.209495>

When you resubmit your manuscript, please download our CHECKLIST

(<http://bit.ly/EMBOPressAuthorChecklist>) and include the completed form in your submission.

Please note that the Author Checklist will be published alongside the paper as part of the transparent process

(<https://www.embopress.org/page/journal/17444292/authorguide#transparentprocess>).

If you feel you can satisfactorily deal with these points and those listed by the referees, you may wish to submit a revised version of your manuscript. Please attach a covering letter giving details of the way in which you have handled each of the points raised by the referees. A revised manuscript will be once again subject to review and you probably understand that we can give you no guarantee at this stage that the eventual outcome will be favorable.

Yours sincerely,

Jingyi Hou

Editor

Molecular Systems Biology

If you do choose to resubmit, please click on the link below to submit the revision online *within 90 days*.

Link Not Available

IMPORTANT: When you send your revision, we will require the following items:

1. the manuscript text in LaTeX, RTF or MS Word format
2. a letter with a detailed description of the changes made in response to the referees. Please specify clearly the exact places in the text (pages and paragraphs) where each change has been made in response to each specific comment given
3. three to four 'bullet points' highlighting the main findings of your study
4. a short 'blurb' text summarizing in two sentences the study (max. 250 characters)
5. a 'thumbnail image' (550px width and max 400px height, Illustrator, PowerPoint or jpeg format), which can be used as 'visual title' for the synopsis section of your paper.
6. Please include an author contributions statement after the Acknowledgements section (see <https://www.embopress.org/page/journal/17444292/authorguide>)
7. Please complete the CHECKLIST available at (<https://bit.ly/EMBOPressAuthorChecklist>). Please note that the Author Checklist will be published alongside the paper as part of the transparent process (<https://www.embopress.org/page/journal/17444292/authorguide#transparentprocess>).
8. Please note that corresponding authors are required to supply an ORCID ID for their name upon submission of a revised manuscript (EMBO Press signed a joint statement to encourage ORCID adoption). (<https://www.embopress.org/page/journal/17444292/authorguide#editorialprocess>)

Currently, our records indicate that there is no ORCID associated with your account.

Please click the link below to provide an ORCID:

Link Not Available

The system will prompt you to fill in your funding and payment information. This will allow Wiley to send you a quote for the article processing charge (APC) in case of acceptance. This quote takes into account any reduction or fee waivers that you may be eligible for. Authors do not need to pay any fees before their manuscript is accepted and transferred to the publisher.

*** PLEASE NOTE *** As part of the EMBO Press transparent editorial process initiative (see our Editorial at <https://dx.doi.org/10.1038/msb.2010.72>), Molecular Systems Biology publishes online a Review Process File with each accepted manuscripts. This file will be published in conjunction with your paper and will include the anonymous referee reports, your point-by-point response and all pertinent correspondence relating to the manuscript. If you do NOT want this File to be published, please inform the editorial office at msb@embo.org within 14 days upon receipt of the present letter.

Reviewer #1:

The authors quantify the effects of ~1.5k temperature sensitive alleles in ~500 yeast essential genes in ten different natural yeast strains. ~180/500 genes had effects suppressed in at least 1 strain. They then use bulk segregant analysis to map modifier loci and find that for many alleles the genetic architecture is simple with one strong modifier gene.

Overall I find this a very interesting study with thought-provoking/provocative extrapolations to human genetic diseases - if similar suppression occurs in humans, then it suggests that many monogenic diseases may be strongly modulated by variation in other loci. The analyses are well performed and the text is easy to follow.

Major suggestion:

The laboratory strain of *S. cerevisiae* is known to have accumulated strong loss-of-function (lof) mutations in several genes, either during laboratory adaptation or neutrally. HAP1 and MKT1 are, I think, examples of such alleles. To what extent does the conclusion that there are individual large suppressor loci for many ts alleles depend on these strong lof alleles present in the laboratory strain? For example, if suppressors are only compared between 'wild' (non lab adapted) strains, is the conclusion similar?

Reviewer #2:

In this work, the authors study the diversity of genetic interactions in 10 natural strains of budding yeast *Saccharomyces cerevisiae*. They use an approach to cross each strain with a laboratory reference strain harboring a temperature-sensitive (ts) allele of an essential gene, and then they score progeny for complementation of the growth defect at the non-permissive temperature. They quantify the various types of interactions, which is interesting. Using the power of yeast genetics, they identify several modifiers and then characterize the types of modifier genes - in most cases, the identified modifiers interact physically with the ts gene or others in the same complex.

The paper is a nice example of how model organisms can be used to quantify these types of interactions and get down to causal alleles in order to study trends in architecture and function. I had several critiques that should be addressed before publication.

First, the methods are vague in places. There is a single reference for the SGA analysis, but it would be useful to at least say some details about what markers are being scored - I believe the ts allele is linked to NAT resistance which is being selected for in all the progeny. But it was unclear to me if single spores are being selected and studied, or if the authors are characterizing pools of spores emerging from a single cross - more details on these methods are required.

If the authors are scoring pools of spores I am concerned how well this approach will work. Each analysis will interrogate different, mixed combination of genotypes. Strong-effect modifiers will be readily identified in this approach, but minor alleles will likely be missed which may be biasing their results. Later in the manuscript, the authors discuss scoring individual spores but cite that wide variation in colony sizes confounds the analysis. Perhaps because I was missing some details, I did not find the first part of the manuscript in which they quantified these effects super convincing. I would have been more excited about the manuscript if I felt these numbers were robust.

I was surprised that for the 154 genes where multiple ts alleles were tested, but only 16% of the multi-allele tests were consistent? This seems exceptionally low. The authors say something about differences in temperature sensitivity, but that didn't make sense to me.

The Methods cite that biological replicates were done for only 6 of 10 strains, and I didn't get a good sense of how reproducible the measurements are. That 4 of the strains had no replication

doesn't seem robust to me.

It is nice that the authors can get down to individual genes, and the fact that many of those they identified participate in direct interactions is interesting.

I did disagree with some of the quantification in the Discussion, including that 78% of the 9 modifiers they identified appear to be gain-of-function alleles ... rather, it seems to me that many of these are more likely reflecting LOSS of function in the lab strain, which is the outlier. This is almost certainly the case for Mkt1, which comes up in just about every QTL analysis involving S288c-derived lab strains and strongly suggests a defective allele in the lab strain. That functional MKT1 can complement numerous alleles that are ts in the lab strain to me suggests that these are not real ts alleles in most strains, but rather that the mkt1 lab background is sensitized to sequence perturbations. The authors do address this later in the Discussion, but it should be addressed in this quantification as well - how many of their "complementations" are rather due to the lab strain being weird and sensitized?

Other minor points:

- The authors mention that they remove 379 34C-temperature-insensitive strains, but it is unclear why - aren't these those that show complete independence of the ts allele? Perhaps I'm misunderstanding, but more clarification would be useful.

- What is the dashed vertical line in Fig 3B?

- The sporulation analysis in Fig S4 would have been much more convincing if done on tetrads rather than 12 randomly selected spores. I wondered if they could really estimate the number of weak-effect alleles from this analysis.

Reviewer #3:

The work by Parts and colleagues addresses an important question of the impact of the genetic background on the manifestation of a loss-of-function allele of an essential gene. To do so, around 1500 temperature sensitive (TS) alleles of essential genes were crossed with 10 natural genetic backgrounds. After excluding controls and alleles that were not TS, the authors observed that 246 of 1,067 TS (23%), corresponding to 35% of tested genes, could be suppressed by some genetic element present in one of the other strains. On average, 1-2 strong modifier loci were identified per cross with a few extra reproducible regions of lower effect size. 102 suppression effects measured in the screen were re-tested for via sporulation with around 50% showing concordance with the screen. From here the authors were able to map some of the modifiers identifying causal genes for 17 candidate loci from 31 attempted, with 9 gene pairs confirmed (TS and modifier). Although the number of confirmed modifiers was small, the evidence so far in the study points to the effect being due to a single or small number of variants of larger effect. Finally, interactions involving the SMC5/6 were studied in some additional detail.

As the authors pointed out in the introduction, other studies have shown that gene deletion phenotypes (including lethality) can depend on the genetic background. So far, the largest studies

have focused on condition dependent growth changes, while this study focuses on lethality making use of the TS alleles. Perhaps the most important advance here is the mapping of the modifiers with the potential important message that few modifier regions of strong effect are sufficient to cause the background dependencies in gene essentiality. These are important findings for human genetic variation although the number of mapped modifiers is still relatively small to make broad conclusions on this.

The work is technically well done and goes through what I would expect for such a study, including the large scale effort, reproducibility of the approach, and findings derived from the mapping. While one could always find ways to continue such a project I find this to be very complete and I have no major concern.

Minor concerns

One prior study that seems very relevant for this manuscript is the work of Mullis and colleagues (Mullis Nat Commun. 2018) where they mapped the modifiers of growth under growth in 10 environments for 7 knock-outs in a BY×3S cross. While it seems difficult to compare that study directly with this one, Mullis and colleagues seem to say that the background effects explaining differences in gene loss of function are highly polygenic and complex. In contrary, the observations from this current study suggest that the modifiers of loss-of-function effects are typically few with strong effects. Given that this point about the number of modifier loci is a critical message of the manuscript, it would be worth to have a better discussion section that integrates the findings of Mullis et al and any such study were the genetic structure of modifiers of loss of function effects has been discussed.

I could not understand the statistics around defining a significant suppression from the screens. Seems that the authors had replicates and could define effect size and variation. There is an FDR estimate a cut-off of 0.75 of suppression but I could not easily figure out how the statistics were done. This needs to be much better explained in the methods section.

I also could not understand exactly how this was done: "As a negative statistical control, a smaller number of 36 out of 530 successfully tested genes (7%) had at least one allele supporting suppression at the permissive temperature"

Given that some strains carry natural variants that can suppress loss of function mutations in specific genes, do the authors find in those same strains predicted loss of function mutations in the corresponding TS genes? Are the genes with TS alleles carrying potential deleterious variants in the strains harbouring the suppressor variants?

Editor's comments

We have made all the changes suggested by the editor, most importantly by providing text and a figure for the synopsis and author contributions, and by replacing the supplementary information with the appendix and expanded view format.

Reviewer comments:

We thank all Reviewers for their positive and constructive comments on our manuscript. Specific comments are addressed below. Changes to the manuscript text are highlighted in yellow in the manuscript file.

In the course of performing re-analyses for the resubmission, we applied additional filters to the considered strains, and slightly modified our overall analysis pipeline, which resulted in minor changes to overall strain counts, and reduced our estimates of suppression frequency (35% to 26% of genes with suppression). Mapping results were unaffected, and there are no changes to the overall conclusions.

Reviewer #1:

The authors quantify the effects of ~1.5k temperature sensitive alleles in ~500 yeast essential genes in ten different natural yeast strains. ~180/500 genes had effects suppressed in at least 1 strain. They then use bulk segregant analysis to map modifier loci and find that for many alleles the genetic architecture is simple with one strong modifier gene.

Overall I find this a very interesting study with thought-provoking/provocative extrapolations to human genetic diseases - if similar suppression occurs in humans, then it suggests that many monogenic diseases may be strongly modulated by variation in other loci. The analyses are well performed and the text is easy to follow.

Major suggestion:

1. The laboratory strain of *S. cerevisiae* is known to have accumulated strong loss-of-function (lof) mutations in several genes, either during laboratory adaptation or neutrally. HAP1 and MKT1 are, I think, examples of such alleles. To what extent does the conclusion that there are individual large suppressor loci for many ts alleles depend on these strong lof alleles present in the laboratory strain? For example, if suppressors are only compared between 'wild' (non lab adapted) strains, is the conclusion similar?

We looked at the TS alleles for which we confirmed that they can be suppressed by "wild" MKT1 alleles, and asked whether these TS alleles are suppressed in all (or most) tested genetic backgrounds, except the S288C control. Although we found examples of this (MED7), it was not always the case. MKT1 alleles differ across strains, and in some cases (GPI13, RPN11), only the MKT1-UWOPS872421 and MKT1-NCYC110 alleles, which both carry additional mutations

(K353E and S371Y+R731G, respectively) that are not present in any of the other tested wild strains, seemed capable of suppressing the TS allele. While we could not identify a study that did not involve the reference strain mapping the MKT1 locus, only a small fraction of genetic diversity in the species has been used for linkage analysis to date, and the non-synonymous variants unique to UWOPS87-2421 and NCYC110 can plausibly have substantial phenotypic effect.

HAP1 may represent a clearer example of a loss-of-function allele in S288C, as it carries a transposon insertion in this background. However, as we had only one example of suppression by HAP1 in our set of validated suppressors, which was also specific to one allele of RPS5 (Fig 4) and not seen for a second rps5 allele, it is hard to draw any conclusions for this gene.

To more systematically look at how often S288C is “the outlier” compared to other yeast strains, we investigated how many TS alleles suppressed by any wild strain were suppressed by

most of them. We relaxed the suppression threshold from 0.75 to 0.5 to avoid winner’s curse and edge effects, and found that this number is moderate, with 61 out of 192 TS alleles (about one third) with substantial suppression in at least 6 wild strains. We have added a new figure panel 2C (copied to the left) and added this analysis to the results section, page 6: “In 31% of the cases, the TS allele is possibly temperature sensitive only in the reference background, with suppression in

segregant progeny of most of the tested wild strains (61 of 192 TS alleles with suppression at a lower 0.5 threshold, e.g. TS alleles of GAB1, Fig 2C and D). For other alleles, suppression was generally limited to a single wild strain (e.g. TS alleles of NSE4, Fig 2D).”

Finally, we re-calculated suppression relative to each of the wild strains as a reference. We observed that the number of TS alleles with suppression values above 0.75 was overall consistent with the phenotype-derived strain tree, and their pattern similar to that of using S288C as a reference (Pearson’s R between the number of TS alleles not in copy number variable regions that are suppressed by each strain at 0.75 cut-off: 0.83 to 0.97). For example, we see that the most genetically diverged strains NCYC110 and UWOPS87-2421 are associated with a high number of suppression events regardless of

the strain used as a reference (new Appendix Fig S3, copied here, colour = number of suppressed TS alleles by a wild strain (x-axis) compared to a given reference strain (y-axis)). Using S288C as reference results in the most suppression events, consistent with additional lab-acquired loss-of-function alleles (top row). Overall, these patterns indicate that it is the combination of the properties of both reference and wild strains that determines the outcomes. We have updated the results section (page 6, Appendix Fig S3) and Discussion (page 15) accordingly.

Reviewer #2:

In this work, the authors study the diversity of genetic interactions in 10 natural strains of budding yeast *Saccharomyces cerevisiae*. They use an approach to cross each strain with a laboratory reference strain harboring a temperature-sensitive (ts) allele of an essential gene, and then they score progeny for complementation of the growth defect at the non-permissive temperature. They quantify the various types of interactions, which is interesting. Using the power of yeast genetics, they identify several modifiers and then characterize the types of modifier genes - in most cases, the identified modifiers interact physically with the ts gene or others in the same complex.

The paper is a nice example of how model organisms can be used to quantify these types of interactions and get down to causal alleles in order to study trends in architecture and function. I had several critiques that should be addressed before publication.

1. First, the methods are vague in places. There is a single reference for the SGA analysis, but it would be useful to at least say some details about what markers are being scored - I believe the ts allele is linked to NAT resistance which is being selected for in all the progeny. But it was unclear to me if single spores are being selected and studied, or if the authors are characterizing pools of spores emerging from a single cross - more details on these methods are required.

We indeed used natMX as one of the selection markers in SGA. We realised that we were inadvertently citing the wrong Baryshnikova et al., 2010 paper in the methods section. We corrected this and are now citing the correct paper, which contains a very detailed SGA protocol. We followed the standard SGA procedure exactly as described in the cited protocol, with the exception of the addition of mannose to the media. At the end of the SGA protocol, each isolated colony represents a pool of individual progeny. We have clarified this both in the main text as well as in the methods.

The results section (page 4) now reads: "We isolated pools of ~60,000 segregant progeny carrying the TS allele to obtain diverse populations of haploid individuals with genomes that, except for the genomic regions around the TS allele and selection markers, are a mosaic of the reference and wild parents (Fig 1B, Methods). We grew the segregant pools at permissive (26°C) and restrictive (34°C) temperatures and measured their fitness."

And the methods (page 18): “SGA analysis was performed as described previously (Baryshnikova et al. 2010), with the exception that 5% mannose was added to the YPD plates used in the first steps of SGA analysis to facilitate pinning of the wild isolates. In brief, the 10 SGA-compatible kanMX-marked wild strains (Dataset EV8, MATa hoΔ::hphMX6 ura3Δ::kanMX4 his3Δ1 leu2Δ0), and a S288C negative control strain (DMA1, MATa his3Δ::kanMX ura3Δ0 leu2Δ0 met15Δ0 or DMA809, MATa hoΔ::kanMX his3Δ0 ura3Δ0 leu2Δ0 met15Δ0; Dataset EV8) were crossed to a collection of 1,474 natMX-marked temperature sensitive mutants of essential genes (MATα xxx-ts::natMX4 can1Δ::STE2pr-SpHIS5 lyp1Δ his3Δ1 leu2Δ0 ura3Δ0 met15Δ0; (Costanzo et al. 2016)). Each cross was performed in four technical replicates, and for six wild strains (Y14273, Y14274, Y14275, Y14276, Y14277, Y14280; see Dataset EV8 for strain information) and the S288C control we performed an additional biological replicate, also containing 4 technical replicates. In a series of subsequent pinning steps, diploid cells were selected and sporulated, and colonies consisting of pools of around 60,000 haploid segregant progeny (Parts et al. 2014) carrying both natMX and kanMX selection markers were isolated. The final selection step for haploid progeny carrying both markers was performed at both 26 and 34°C.”

2. If the authors are scoring pools of spores I am concerned how well this approach will work. Each analysis will interrogate different, mixed combination of genotypes. Strong-effect modifiers will be readily identified in this approach, but minor alleles will likely be missed which may be biasing their results. Later in the manuscript, the authors discuss scoring individual spores but cite that wide variation in colony sizes confounds the analysis. Perhaps because I was missing some details, I did not find the first part of the manuscript in which they quantified these effects super convincing. I would have been more excited about the manuscript if I felt these numbers were robust.

We are indeed using pools of progeny. Each population in our screen initially consists of 60,000 different viable haploid progeny that will grow up to form a single colony of ~20 million cells (Parts et al., 2014, Genome Res). Although each population (colony) isolated from the same cross will be slightly different, the number of individuals is sufficient to generate reproducible results. Indeed, the fitnesses at the restrictive temperature of 34C are very consistent between biological replicates (Pearson’s R 0.90, 0.89, 0.88, 0.87, 0.81, 0.72 for the six wild strains with biological replicates; median of 0.86; Pearson’s R=0.82 for the control; updated Appendix Figure S1).

Each population is grown for ~30 generations. This number of generations, in combination with the number of cells per population, should be large enough to detect relatively weak modifiers. Indeed, we do see weaker modifiers (31 strong suppressors with allele frequency change of at least 0.2, but an additional 48 ones with allele frequency change between 0.14 and 0.2) that are also reproducible (Fig 3C).

Almost two-thirds of the suppression events we measured at scale confirmed in a follow-up experiment (62%). We believe the reasons for not calling the remaining fraction reproduced are multiple. First, we use stringent thresholds for calling a TS allele suppressed in our validation assay. In this assay, we examined the growth of hundreds of individual progeny from each cross. As there are tens of thousands of variants segregating in each population, the

progeny of most crosses showed high variation in colony size, which was further influenced by the number of colonies on the plate, complicating the identification of differences in growth compared to a control cross. Indeed, we could increase the random sporulation based follow-up reproducibility rate by using a computational threshold of two-fold increase in colony size, rather than visual inspection (lower y-axis cut-off in Fig EV2E), however, this computational threshold increased the number of false positive calls. Second, in the screen we used populations of millions of cells, whereas we only examined a few hundred cells in our validation assay. Rare mechanisms of suppression, such as sporadic aneuploidisation to escape selection, are thus another possible source of false positive screen suppression signal. In spite of these issues, we observed high correlation between different TS alleles of the same gene and independent replicates of our screen (see point #3 and #4 below), showing that the vast majority of suppression events we detected are real. We have added this discussion to the text (page 14).

3. I was surprised that for the 154 genes where multiple ts alleles were tested, but only 16% of the multi-allele tests were consistent? This seems exceptionally low. The authors say something about differences in temperature sensitivity, but that didn't make sense to me.

We apologise for the lack of clarity. Of all the genes with multiple alleles, 16% showed suppression of two or more alleles by some wild strain variants. This number is low, because the vast majority of genes are never suppressed. In other words, 16% of ALL genes with multiple TS alleles showed suppression supported by a second allele, not 16% of suppressed genes. We have now removed this sentence from the text. Instead, we report that when we observe suppression for a combination of a TS allele and wild strain, the strongest suppression of another TS allele of the same gene is substantially larger than that of a randomly picked TS allele on average (0.61 vs 0.39). We have added this claim to the main text (page 5) and its visualisation to the new Appendix Figure S2. Please see the previous and next points for further details on reproducibility.

4. The Methods cite that biological replicates were done for only 6 of 10 strains, and I didn't get a good sense of how reproducible the measurements are. That 4 of the strains had no replication doesn't seem robust to me.

We have added biological replicate reproducibility scatter plots at permissive and restrictive temperatures to Appendix Figure S1, and now also report technical reproducibility in Dataset EV1. Most large scale SGA screens employ only technical replicates located on the same agar plate (Costanzo et al., 2016, Science; Baryshnikova et al., 2010, Nat Methods). We tested whether suppression values derived from technical replicates only are systematically more variable, and thus lead to increased discoveries due to measurement noise, compared to ones derived from technical and biological replicates. To do so, we evaluated whether the variability of reported fitness values, or the number of called suppression events, depends on the number of biological replicates. We considered the Wine/European strains as the nominally most similar, leaving out the Y14276 phenotypic outlier. This set is comprised of three strains with two biological replicates, and three strains with one biological replicate. We calculated variability of

suppression values within both sets, and found that for 657 of 1073 TS alleles (after filtering for ones measured in all strains, and not in translocated region), the wild strains with two biological replicates had higher variance across strains than those with a single replicate. Concordantly, the strains with a single replicate had on average 36 alleles suppressed, while the strains with two replicates had 35 alleles suppressed. Together, these results argue against a substantial difference in discoveries due to the number of biological replicates. Finally, in our experiment, each “technical” replicate consists of a unique, independently generated pool of 60,000 segregants. Our technical replicates thus also cover some biological variation.

5. It is nice that the authors can get down to individual genes, and the fact that many of those they identified participate in direct interactions is interesting.

We thank the reviewer for this comment, and share their enthusiasm.

6. I did disagree with some of the quantification in the Discussion, including that 78% of the 9 modifiers they identified appear to be gain-of-function alleles ... rather, it seems to me that many of these are more likely reflecting LOSS of function in the lab strain, which is the outlier. This is almost certainly the case for Mkt1, which comes up in just about every QTL analysis involving S288c-derived lab strains and strongly suggests a defective allele in the lab strain. That functional MKT1 can complement numerous alleles that are ts in the lab strain to me suggests that these are not real ts alleles in most strains, but rather that the mkt1 lab background is sensitized to sequence perturbations. The authors do address this later in the Discussion, but it should be addressed in this quantification as well - how many of their "complementations" are rather due to the lab strain being weird and sensitized?

Please see reviewer 1 point 1 for a discussion on the lab reference strain being sensitised compared to other yeast strains. Indeed, a substantial fraction (about one third) of the suppression events are consistent with a loss-of-function event in the reference strain. We thank the reviewer for this suggestion, and have added a new figure panel 2C (copied to the right) to highlight this observation.

Regarding the percentage of suppressors with a gain-of-function effect, we have now included the calculation without MKT1 alleles as well (page 11): “When we exclude MKT1, 3 out of 5 (60%) suppressor genes were classified as gain-of-function alleles compared to the reference allele.”

We also added the following text to the discussion (page 15): “Alternatively, the high fraction of gain-of-function effects among natural suppressors may result from a loss-of-function defect of the gene in the reference background. This is likely true for HAP1 and MKT1 alleles (see below). However, we found no evidence for this being the dominant mechanism, as two

thirds of the suppressed genes showed suppression by natural variation in only a small subset of the wild backgrounds (Fig 2C)."

Other minor points:

7. The authors mention that they remove 379 34C-temperature-insensitive strains, but it is unclear why - aren't these those that show complete independence of the ts allele? Perhaps I'm misunderstanding, but more clarification would be useful.

These 379 strains (292 in our updated analysis) did not show a growth defect at 34C in the reference (S288c) background. Because these TS alleles show WT-growth at 34C in the reference strain, we were unable to determine whether they could be suppressed. Most likely, these TS alleles have a restrictive temperature that is substantially higher than 34C, and would need to be queried at a higher temperature. To prevent confusion, we removed these alleles from the main text, and only describe them in the methods section where we use them as calibration controls. As a result, we replaced the number of tested alleles from 1498 to 1106 everywhere in the main text. In the methods section (page 19), we changed the text to: "We filtered out 76 strains that were either missing in the reference cross or all other samples at the restrictive temperature after filtering. We further filtered out 292 temperature-insensitive query strains that did not show lower fitness at the restrictive temperature in the reference strain background (fitness difference between relative colony sizes at 26 and 34°C below 0.2), retaining 1,106 query strains in total for 580 genes. "

8. What is the dashed vertical line in Fig 3B?

This is the cut-off we used for calling QTLs. We added the following sentence to the figure legend: "Vertical line at an allele frequency change of 0.2: the cut-off we used for calling suppressor loci."

9. The sporulation analysis in Fig S4 would have been much more convincing if done on tetrads rather than 12 randomly selected spores. I wondered if they could really estimate the number of weak-effect alleles from this analysis.

We agree, however, there are a few reasons why we used a modification of the "standard" tetrad-based analysis. First, several of these crosses had a reduced spore viability, independent of the TS allele. For example, several strains carry a translocation between chrVIII and chrXVI, which leads to 25% spore lethality (see also Fig. EV1). When dissected at 34C, spores carrying the TS allele can thus die either because of lack of a modifier, or because of lack of part of chrXVI due to the translocation, which complicates interpretation of the segregation patterns. To circumvent this, we dissected the strains at 26C, a temperature at which spore lethality should not be related to the TS allele. We tried replica plating these dissection plates to 34C to determine growth differences at the restrictive temperature, but this was difficult to see on the replica plates. Instead, we decided to pick the first 12 spores carrying the TS allele from the dissection plate, and tested their growth using the spot dilutions shown in Fig S4 (now called Fig EV3). We agree that this number of spores and analysis is not sufficient to determine the

number of weak-effect alleles. We changed the text of the figure legend to reflect this: "Estimating the number of strong modifiers. TS alleles were crossed to the wild strain in which they were suppressed, and the resulting hybrid strains were dissected at 26C. The first 12 spores carrying the TS allele were selected from the dissection plate, and were grown overnight in liquid media. Cultures were diluted to an optical density at 600 nm of 0.1 and a series of ten-fold dilutions was spotted on agar plates and incubated for 2 days at 26C or 34C. The number of strong modifiers was estimated by determining the fraction of the spores that grew well at 34C."

Reviewer #3:

The work by Parts and colleagues addresses an important question of the impact of the genetic background on the manifestation of a loss-of-function allele of an essential gene. To do so, around 1500 temperature sensitive (TS) alleles of essential genes were crossed with 10 natural genetic backgrounds. After excluding controls and alleles that were not TS, the authors observed that 246 of 1,067 TS (23%), corresponding to 35% of tested genes, could be suppressed by some genetic element present in one of the other strains. On average, 1-2 strong modifier loci were identified per cross with a few extra reproducible regions of lower effect size. 102 suppression effects measured in the screen were re-tested for via sporulation with around 50% showing concordance with the screen. From here the authors were able to map some of the modifiers identifying causal genes for 17 candidate loci from 31 attempted, with 9 gene pairs confirmed (TS and modifier). Although the number of confirmed modifiers was small, the evidence so far in the study points to the effect being due to a single or small number of variants of larger effect. Finally, interactions involving the SMC5/6 were studied in some additional detail.

As the authors pointed out in the introduction, other studies have shown that gene deletion phenotypes (including lethality) can depend on the genetic background. So far, the largest studies have focused on condition dependent growth changes, while this study focuses on lethality making use of the TS alleles. Perhaps the most important advance here is the mapping of the modifiers with the potential important message that few modifier regions of strong effect are sufficient to cause the background dependencies in gene essentiality. These are important findings for human genetic variation although the number of mapped modifiers is still relatively small to make broad conclusions on this.

The work is technically well done and goes through what I would expect for such a study, including the large scale effort, reproducibility of the approach, and findings derived from the mapping. While one could always find ways to continue such a project I find this to be very complete and I have no major concern.

Minor concerns

1. One prior study that seems very relevant for this manuscript is the work of Mullis and colleagues (Mullis Nat Commun. 2018) where they mapped the modifiers of growth

under growth in 10 environments for 7 knock-outs in a BY×3S cross. While it seems difficult to compare that study directly with this one, Mullis and colleagues seem to say that the background effects explaining differences in gene loss of function are highly polygenic and complex. In contrary, the observations from this current study suggest that the modifiers of loss-of-function effects are typically few with strong effects. Given that this point about the number of modifier loci is a critical message of the manuscript, it would be worth to have a better discussion section that integrates the findings of Mullis et al and any such study were the genetic structure of modifiers of loss of function effects has been discussed.

We thank the Reviewer for bringing this paper to our attention, and apologise for not including it earlier. Mullis et al identified complex background-dependence of knock-out mutation effects in a panel of unselected segregants, with 15 to 350 linkages per environment. We found that a strong linkage to a suppressor exists in every selected population we sequenced. We believe that these two sets of results are consistent with a view where there is complex background dependency of many mapped loci, which is sometimes (at most 4% of allele-strain combinations in our data) peppered with large effect modifier alleles.

The apparent discrepancies arise due to our selection of crosses to use for mapping, and the statistics we emphasise in the text. First, we mostly picked suppression events where the phenotypic advantage was large, thus also implicitly biasing the selected populations for large genetic effects. Mullis et al performed their mapping in an unbiased way with seven knock-out strains, for which there plausibly were no large effects to map, while we could select from over a thousand partial loss-of-function alleles. Second, while we report the strongest genetic effects, there are additional weaker ones that are reproducible (Figures 3B and 4A). We did not focus on these as they are very challenging to finemap and confirm, but the genetic complexity likely exists in our linkage maps as well, as for nearly all traits.

While we believe the results of the studies are consistent, we choose to emphasise the (to us) more surprising finding that a single large effect allele was sufficient to rescue growth in all validated cases. We have now included a reference to the Mullis paper in the introduction (page 3), and expanded the discussion on genetic complexity to include the arguments above, that also includes a reference to this paper (page 13).

2. I could not understand the statistics around defining a significant suppression from the screens. Seems that the authors had replicates and could define effect size and variation. There is an FDR estimate a cut-off of 0.75 of suppression but I could not easily figure out how the statistics were done. This needs to be much better explained in the methods section.

Briefly, we treated the measurements as emissions from normal distributions with unknown mean and variance, inferred the posteriors for growth and growth differences from the data, and use their mean to give estimates of suppression, and the area under the tail to calculate the frequency at which we expect the value to be more extreme. We report the latter as the p-value.

We have added the following text to the methods section (page 19): “To estimate suppression of the mutation effect by a wild strain, we quantified the difference in fitness

(normalised log₂-scale colony size) at the restrictive temperature after adjusting for overall growth differences between the reference and wild strains. To adjust for global growth differences, we set the median restrictive temperature fitnesses of temperature-insensitive strains (see previous section) to be equal, and scaled the wild strain restrictive temperature fitnesses to minimise mean-squared error of the fit to the respective values of reference. To obtain posterior variance estimates of suppression, we used the posterior normal distributions of the fitnesses of the wild and reference strain crosses to obtain the posterior distribution for the difference, treating them as independent (thus adding variances), and computed its mean and standard deviation. We also calculated z-scores of suppression (for the mean suppression to be above 0) as mean divided by standard deviation. We called a TS allele suppressed if the mean adjusted fitness in the wild strain cross was at least 0.75 larger at 34°C than the S288C reference (i.e., colonies on average 1.68 times bigger), and the z-score was at least 4.5. We decided on a cut-off of 0.75 as this corresponds to a visually clear difference. The z-score of 4.5 corresponds to a nominal one-sided p-value of 3.4×10^{-6} for a normal distribution, and a Bonferroni-corrected p-value of 0.0359 after adjusting for 10,554 total tests corresponding to 10 wild strains crossed to the 1,106 queries, after filtering out missing values.”. The code that does the calculation is available on FigShare for reproducibility and inspection (<http://dx.doi.org/10.6084/m9.figshare.14170787>).

Please also see reviewer #2 point #4 on replicate reproducibility.

3. I also could not understand exactly how this was done: "As a negative statistical control, a smaller number of 36 out of 530 successfully tested genes (7%) had at least one allele supporting suppression at the permissive temperature"

At the permissive temperature (26C), we do not expect to see a large amount of suppression, because most (but not all) TS alleles can support normal growth at this temperature. The fact that we saw little suppression at 26C (7% of tested genes) compared to 34C (35% of tested genes in the previous submission) is thus concordant with the expectation, and speaks to the quality of our screens. However, we removed this sentence, as well as the suppression data on temperature-insensitive strains, from the manuscript, as we already describe several other quality control measures, and this particular sentence may mainly cause confusion without adding a lot of information.

4. Given that some strains carry natural variants that can suppress loss of function mutations in specific genes, do the authors find in those same strains predicted loss of function mutations in the corresponding TS genes ? Are the genes with TS alleles carrying potential deleterious variants in the strains harbouring the suppressor variants ?

We compared the presence and number of deleterious variants (determined by SIFT, Ng and Henikoff (2001, Genome Res), downloaded from SGRP2 project website <http://www.moseslab.csb.utoronto.ca/sgrp/download.html>) in a gene in genetic backgrounds in which TS alleles of that gene were suppressed to genetic backgrounds in which it was not suppressed. We found no convincing evidence for the accumulation of deleterious mutations in

genetic backgrounds in which the gene could be suppressed (e.g. average number of deleterious mutations in suppressable genes 0.14 vs 0.16 in the rest).

The reasons for this could be plentiful. Because we only measured suppression at 34C, and some TS alleles have restrictive temperatures that are significantly lower or higher, we may have classified some genes as “not suppressed” even though they could have been suppressed if measured at a different temperature. Alternatively, maybe the definition of “deleterious” we used here was too extreme. The identified suppressors most likely depend on the presence of the TS allele, and thus can only suppress partial loss-of-function events, and not full loss-of-function alleles (see for example Fig. EV5B). However, when comparing distributions of SIFT scores between suppressed and non-suppressed cases, the differences remained negligible. We now briefly describe these results on page 7.

16th Apr 2021

Manuscript Number: MSB-2020-10138R

Title: Natural variants suppress mutations in hundreds of essential genes

Author: Leopold Parts

Amandine Batte

Maykel Lopes

Michael Yuen

Meredith Laver

Bryan-Joseph San Luis

Jia-Xing Yue

Carles Pons

Elise Eray

Patrick Aloy

Gianni Liti

Jolanda Van Leeuwen

Dear Dr. Parts,

Thank you for sending us your revised manuscript. We have now heard back from the three reviewers who were asked to evaluate your study. As you will see, the reviewers are satisfied with the modifications made and think that the study is now suitable for publication.

Before we can formally accept your manuscript, we would ask you to address the following editorial-level issues.

1. Please provide five keywords and incorporate them in the main text.
2. Synopsis image: the text becomes too small and somewhat blurry (see attached for the resized image) when the synopsis image is adjusted to the required size (550 px width). Please provide a new image (JPEG format) with clearer text (for instance, with increased text size).
3. Data availability: We usually ask authors to deposit only unstructured data to Figshare. Can you please deposit your computer code to a database for structured data, such as Github?
4. Our data editors have seen the manuscript, and they have made some comments and suggestions that need to be addressed (see attached). Please send back a revised version (in track change mode), as we will need to go through the changes.
5. I have slightly modified and shortened the synopsis text since it allows no more than 250 characters, including space. Please let me know if you are fine with it or if you would like to introduce further modifications.

A survey of ~1,100 temperature-sensitive alleles of yeast essential genes in ten diverse strains shows that natural genetic variants frequently suppress deleterious mutations. Genetic mapping and allele replacement identify causal suppressor genes.

- 149 out of 580 tested genes can be suppressed by natural variants from at least one genetic background.
- A single strong suppressor allele can independently overcome the temperature sensitivity phenotype in nearly all mapped cases.
- Identified suppressor genes include members of the same complex or pathway as specific temperature-sensitive mutants, as well as general modifiers that suppress many mutant alleles.

When you resubmit your manuscript, please download our CHECKLIST (<https://bit.ly/EMBOPressAuthorChecklist>) and include the completed form in your submission. *Please note* that the Author Checklist will be published alongside the paper as part of the transparent process (<https://www.embopress.org/page/journal/17444292/authorguide#transparentprocess>)

Click on the link below to submit your revised paper.

I look forward to receiving your revised manuscript as soon as possible.

Kind regards,
Jingyi

Jingyi Hou
Editor
Molecular Systems Biology

If you do choose to resubmit, please click on the link below to submit the revision online before 16th May 2021.

Link Not Available

IMPORTANT: When you send your revision, we will require the following items:

1. the manuscript text in LaTeX, RTF or MS Word format
2. a letter with a detailed description of the changes made in response to the referees. Please specify clearly the exact places in the text (pages and paragraphs) where each change has been made in response to each specific comment given
3. three to four 'bullet points' highlighting the main findings of your study
4. a short 'blurb' text summarizing in two sentences the study (max. 250 characters)
5. a 'thumbnail image' (550px width and max 400px height, Illustrator, PowerPoint or jpeg format), which can be used as 'visual title' for the synopsis section of your paper.
6. Please include an author contributions statement after the Acknowledgements section (see <https://www.embopress.org/page/journal/17444292/authorguide#manuscriptpreparation>)
7. Please complete the CHECKLIST available at (<https://bit.ly/EMBOPressAuthorChecklist>). Please note that the Author Checklist will be published alongside the paper as part of the

transparent process

(<https://www.embopress.org/page/journal/17444292/authorguide#transparentprocess>).

8. Please note that corresponding authors are required to supply an ORCID ID for their name upon submission of a revised manuscript (EMBO Press signed a joint statement to encourage ORCID adoption) (<https://www.embopress.org/page/journal/17444292/authorguide#editorialprocess>).

Currently, our records indicate that the ORCID for your account is 0000-0002-2618-670X.

Link Not Available

The system will prompt you to fill in your funding and payment information. This will allow Wiley to send you a quote for the article processing charge (APC) in case of acceptance. This quote takes into account any reduction or fee waivers that you may be eligible for. Authors do not need to pay any fees before their manuscript is accepted and transferred to the publisher.

*** PLEASE NOTE *** As part of the EMBO Press transparent editorial process initiative (see our Editorial at <https://dx.doi.org/10.1038/msb.2010.72> , Molecular Systems Biology will publish online a Review Process File to accompany accepted manuscripts. When preparing your letter of response, please be aware that in the event of acceptance, your cover letter/point-by-point document will be included as part of this File, which will be available to the scientific community. More information about this initiative is available in our Instructions to Authors. If you have any questions about this initiative, please contact the editorial office (msb@embo.org).

Reviewer #1:

The authors have addressed my concerns.

Reviewer #2:

The authors have done a nice job addressing all of my concerns. The manuscript is now clearer with regards to methods and quantification, and the quantification of lab-strain-specific TS alleles is an important addition. I am satisfied with all of their responses.

Reviewer #3:

The authors have addressed all of my previous concerns. I think this adds further evidence of the importance of standing variation with strong impact and effect sizes which is timely given the push to perform larger exome/genome sequencing efforts in human cohorts.

The authors performed the requested editorial changes.

23rd Apr 2021

Manuscript number: MSB-2020-10138RR

Title: Natural variants suppress mutations in hundreds of essential genes

Dear Dr. Parts,

Thank you again for sending us your revised manuscript. We are now satisfied with the modifications made and I am pleased to inform you that your paper has been accepted for publication.

*** PLEASE NOTE *** As part of the EMBO Publications transparent editorial process initiative (see our Editorial at <https://dx.doi.org/10.1038/msb.2010.72>), Molecular Systems Biology publishes online a Review Process File with each accepted manuscripts. This file will be published in conjunction with your paper and will include the anonymous referee reports, your point- by-point response and all pertinent correspondence relating to the manuscript. If you do NOT want this File to be published, please inform the editorial office at msb@embo.org within 14 days upon receipt of the present letter.

Should you be planning a Press Release on your article, please get in contact with msb@wiley.com as early as possible, in order to coordinate publication and release dates.

LICENSE AND PAYMENT:

All articles published in Molecular Systems Biology are fully open access: immediately and freely available to read, download and share.

Molecular Systems Biology charges an article processing charge (APC) to cover the publication costs. You, as the corresponding author for this manuscript, should have already received a quote with the article processing fee separately.

Please let us know in case this quote has not been received.

Once your article is at Wiley for editorial production you will receive an email from Wiley's Author Services system, which will ask you to log in and will present you with the publication license form for completion. Within the same system the publication fee can be paid by credit card, an invoice or pro forma can be requested.

Payment of the publication charge and the signed Open Access Agreement form must be received before the article can be published online.

Molecular Systems Biology articles are published under the Creative Commons licence CC BY, which facilitates the sharing of scientific information by reducing legal barriers, while mandating attribution of the source in accordance to standard scholarly practice.

Proofs will be forwarded to you within the next 2-3 weeks.

Thank you very much for submitting your work to Molecular Systems Biology.

Kind regards,

Jingyi

Jingyi Hou
Editor
Molecular Systems Biology

Corresponding Author Name: Jolanda van Leeuwen, Leopold Parts

Manuscript Number: MSB-2020-10138R